# Multiple motors cooperate to establish and maintain acentrosomal spindle bipolarity in *C. elegans* oocyte meiosis

Gabriel Cavin-Meza, Michelle M Kwan, Sarah M Wignall*

Department of Molecular Biosciences, Northwestern University, Evanston, United States

**Abstract** While centrosomes organize spindle poles during mitosis, oocyte meiosis can occur in their absence. Spindles in human oocytes frequently fail to maintain bipolarity and consequently undergo chromosome segregation errors, making it important to understand the mechanisms that promote acentrosomal spindle stability. To this end, we have optimized the auxin-inducible degron system in *Caenorhabditis elegans* to remove the factors from pre-formed oocyte spindles within minutes and assess the effects on spindle structure. This approach revealed that dynein is required to maintain the integrity of acentrosomal poles; removal of dynein from bipolar spindles caused pole splaying, and when coupled with a monopolar spindle induced by depletion of the kinesin-12 motor KLP-18, dynein depletion led to a complete dissolution of the monopole. Surprisingly, we went on to discover that following monopole disruption, individual chromosomes were able to reorganize local microtubules and re-establish a miniature bipolar spindle that mediated chromosome segregation. This revealed the existence of redundant microtubule sorting forces that are undetectable when KLP-18 and dynein are active. We found that the kinesin-5 family motor BMK-1 provides this force, uncovering the first evidence that kinesin-5 contributes to *C. elegans* meiotic spindle organization. Altogether, our studies have revealed how multiple motors are working synchronously to establish and maintain bipolarity in the absence of centrosomes.

**\*For correspondence:**
s-wignall@northwestern.edu

**Competing interest:** The authors declare that no competing interests exist.

## Editor's evaluation

Oocytes in most species contain acentrosomal spindles and how these are formed and maintained is not entirely clear. Cavin-Meza et al. find that dynein is required for the integrity of bipolar spindle poles as well as spindle monopoles induced by depletion of the kinesin-12 motor KLP-18. They further show that in the absence of dynein motors, and functional spindle poles, microtubules are still able to organize into spindle-like structures via the action of chromosomes and the kinesin-5 motor BMK-1. They also found that chromosomes can still be segregated in an anaphase-like motion in these oocytes.

## Introduction

Centrosomes serve as prominent microtubule organizing centers (MTOCs) during mitosis and provide a clear structural blueprint for the formation of a bipolar spindle. However, oocytes of many species lack centrosomes. In human oocytes, it has been shown that acentrosomal spindles are sometimes highly unstable; poles go through an extended period where they can split apart and then come back together, and spindles that display this instability have a high incidence of chromosome segregation errors (*Holubcová et al., 2015*). The causes of this instability and the mechanisms by which acentrosomal spindles are stabilized in the absence of centrosomes remain poorly understood.

**eLife digest** Meiosis is a specialized form of cell division that produces the gametes required for sexual reproduction, such as egg and sperm cells. Before the cell splits, it copies its genome so that it has four sets of chromosomes. Genetic information is then shuffled between the chromosomes, and the cell undergoes two rounds of division, resulting in four gametes that are genetically distinct.

Prior to division, the duplicated chromosomes are separated by rope-like protein polymers called microtubules. In most cells, structures called centrosomes organize these fibers into a spindle shape that emanates from two 'poles' on opposite ends of the cell: the microtubules then attach to the chromosomes and pull them apart. Despite not having centrosomes, egg cells, or 'oocytes', are still able to arrange their microtubules into a similar bipolar shape. However, how oocytes form these 'acentrosomal' spindles is poorly understood.

Centrosomes do not organize the spindle alone, and receive help from various motor proteins such as dynein. Previous work showed that dynein is involved in arranging acentrosomal poles, but it was not known if it was required to hold the poles together after they initially formed. To investigate, Cavin-Meza et al. developed a strategy that can rapidly remove dynein from oocytes of the round-worm *Caenorhabditis elegans*.

The experiment showed that dynein is required both to assemble and stabilize acentrosomal spindles in *C. elegans*. When dynein and an additional motor protein, KLP-18, were both removed from oocytes simultaneously, the poles blew apart, completely disrupting spindle organization. Surprisingly, Cavin-Meza et al. found that the spindles were able to reform and separate the chromosomes. Further probing revealed, for the first time, that a third motor protein (called BMK-1) also helps to organize the spindle into its bipolar structure.

These findings reveal the important role motor proteins play in stabilizing spindles and separating chromosomes in oocytes. Meiosis is prone to mistakes, and these errors are a major cause of miscarriages and birth defects in humans. Therefore, understanding the underlying mechanisms of how oocyte spindles form and remain stable could shed light on why chromosomes sometimes fail to segregate. This may eventually lead to new strategies for combating infertility.

We use *Caenorhabditis elegans* as a model to identify factors required for acentrosomal pole stability. In *C. elegans* oocytes, microtubules are nucleated in proximity to the disassembling nuclear envelope in a spherical cage-like structure. Microtubule minus ends are then pushed outward to the periphery of the array by the kinesin-12 family motor KLP-18, where they form multiple poles with clearly defined clusters of minus ends. Spindle poles then coalesce, eventually forming a bipolar structure with aligned chromosomes (*Wolff et al., 2016*; *Gigant et al., 2017*). Shortly after establishing a metaphase plate, the spindle shortens and rotates perpendicular to the cortex and chromosomes begin to segregate in anaphase (*Albertson and Thomson, 1993*). The cortical set of homologous chromosomes are discarded as a polar body, and the remaining chromosomes undergo a second round of meiosis to generate the final set of maternal DNA.

While these stages of acentrosomal spindle assembly have been documented in multiple studies, the mechanisms by which microtubules organize into stable acentrosomal poles in this organism are not well understood. It would stand to reason that mitotic pole proteins are ideal candidates for also promoting meiotic pole stability; many proteins that serve important functions in *C. elegans* mitotic spindle assembly can be found within acentrosomal meiotic spindles (reviewed in *Severson et al., 2016*; *Mullen et al., 2019*). Various microtubule-associated proteins (MAPs) and microtubule motors have been demonstrated to concentrate at poles and contribute to acentrosomal spindle assembly, such as ASPM-1 (*Connolly et al., 2014*), KLP-7[MCAK] (*Connolly et al., 2015*; *Han et al., 2015*; *Gigant et al., 2017*), and MEI-1/2[katanin] (*Srayko et al., 2000*). A study using a fast-acting temperature-sensitive *mei-1* mutant demonstrated that katanin is essential to maintain spindle structure (*McNally et al., 2014*), but whether any other pole-associated-proteins are required for spindle maintenance is not known.

Another factor that has been implicated in spindle pole organization is the minus-end-directed microtubule motor dynein. Studies of mitotically dividing cells in multiple organisms have shown that dynein is required for spindle pole focusing and for centrosome anchoring to poles (reviewed in

*Borgal and Wakefield, 2018*) and dynein also promotes poleward flux of newly formed microtubule minus ends (*Elting et al., 2014*; *Hueschen et al., 2017*). Moreover, there is evidence that dynein is involved in the organization of acentrosomal poles. Inhibition of dynein in *Xenopus* egg extracts creates broad, splayed acentrosomal spindle poles (*Heald et al., 1996*), and injection of dynein anti-bodies into *Xenopus* oocytes disrupts overall meiotic spindle organization (*Becker et al., 2003*). In mouse oocytes, dynein is involved in the early stages of spindle assembly, so dynein inhibition prevents spindle (and thus pole) formation entirely (*Luksza et al., 2013*; *Clift and Schuh, 2015*).

In *C. elegans*, dynein is crucial for the proper positioning of both mitotic and meiotic spindles, and for spindle rotation during meiotic anaphase (*Ellefson and McNally, 2009*; *van der Voet et al., 2009*; *Ellefson and McNally, 2011*; *Crowder et al., 2015*). Moreover, a number of studies have shown that depletion of dynein or its cofactor dynactin causes spindle defects in oocytes, such as longer spindles with pole defects (*Yang et al., 2005*; *Ellefson and McNally, 2009*; *Ellefson and McNally, 2011*; *Crowder et al., 2015*; *Muscat et al., 2015*). However, whether dynein is required to maintain acentrosomal pole integrity after bipolarity has been established is not known. Thus, there is a need for an approach that can rapidly and efficiently remove dynein function from oocytes in a conditional manner.

Here, we have addressed this challenge using the auxin-inducible degron (AID) system (*Zhang et al., 2015*). We found that this method can efficiently remove dynein from oocytes within minutes, enabling us to remove this protein either prior to spindle assembly or after bipolarity has already been established. This approach revealed that dynein is essential for both the formation and stabilization of acentrosomal spindle poles. Removing dynein from monopolar spindles caused catastrophic break-down of the monopole, further supporting a role for dynein in pole integrity. Moreover, this experiment also revealed a striking phenotype that led to additional insights into acentrosomal spindle function. Specifically, we found that following disruption of the monopole, individual chromosomes were able to reorganize local microtubules into a bipolar spindle that could mediate anaphase-like segregation. This phenotype provided an opportunity to identify proteins contributing to spindle bipolarity that are masked under normal conditions. Through this assay, we identified the kinesin-5 family motor BMK-1 as an outward sorting force on microtubules, providing the first evidence that kinesin-5 contributes to spindle assembly in *C. elegans*. Altogether, these studies have furthered our understanding of the motor forces that contribute to acentrosomal spindle assembly and maintenance during meiosis.

## Results

### Dynein is localized to acentrosomal spindles and is required for pole focusing

Since dynein is required for multiple cellular processes in *C. elegans*, full RNAi-mediated depletion of this protein causes developmental defects. Thus, previous studies of dynein in oocytes have relied on partial depletion (*Yang et al., 2005*; *Ellefson and McNally, 2009*; *Ellefson and McNally, 2011*; *Muscat et al., 2015*; *McNally et al., 2016*; *Laband et al., 2017*). In an attempt to achieve more complete depletion without affecting worm development, we sought to rapidly deplete dynein from adult worms using the AID system. For this, we utilized a strain in which the heavy chain of dynein, DHC-1, was tagged at the endogenous locus with both GFP and a degron tag; this strain also expressed the ubiquitin ligase TIR1 using the *sun-1* promoter to enable auxin-mediated degra-dation of DHC-1 specifically in the germline (*Zhang et al., 2015*; *Figure 1A*; we refer to this strain as 'Dynein AID'). This strain appears superficially wild type, with no noticeable effects on worm devel-opment, brood size, or percent of male progeny, suggesting that the GFP and degron tags are not affecting dynein function in the absence of auxin (*Zhang et al., 2015*). With this system, we are able to deplete dynein on a shorter timescale than RNAi, allowing us to assess the effects of dynein deple-tion on spindle assembly and maintenance without affecting prior processes. To validate that this strain worked for efficient DHC-1 depletion, we first performed 'long-term' auxin-mediated depletion, growing adult worms on auxin-containing plates for 4 hr (*Figure 1A*). We then imaged one-cell-stage mitotic embryos using immunofluorescence (IF) and looked for canonical dynein depletion pheno-types, such as failure to separate centrosomes and improper mitotic spindle positioning (*Gönczy et al., 1999*; *Nguyen-Ngoc et al., 2007*; *Kiyomitsu and Cheeseman, 2013*). After a 4 hr incubation

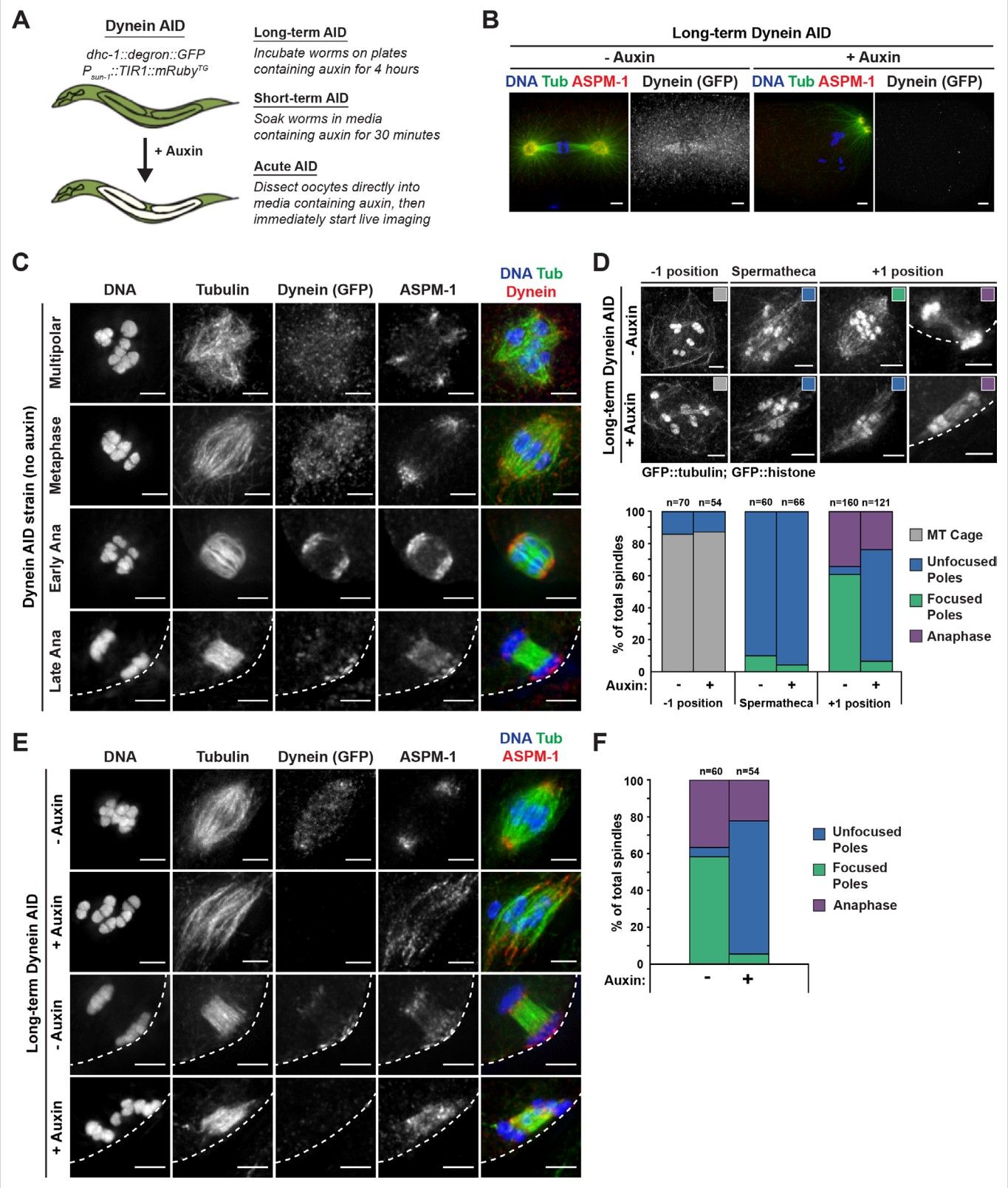

**Figure 1.** Dynein is required for acentrosomal pole focusing. (**A**) Schematic representation of the Dynein auxin-inducible degron (AID) system for DHC-1 depletion in the *C. elegans* germ line. Methodologies for all three auxin treatments used in this study are depicted. (**B**) Immunofluorescence (IF) imaging of one-cell mitotically dividing embryos shows that auxin treatment causes efficient dynein depletion and canonical mitotic spindle defects (n = 22). Shown are tubulin (green), DNA (blue), ASPM-1 (red), and dynein (not shown in merge). (**C**) IF imaging of oocyte spindles in the

*Figure 1 continued on next page*

*Figure 1 continued*

Dynein AID strain shows that dynein is localized to the spindle, with increasing enrichment at acentrosomal poles at the anaphase transition; shown are tubulin (green), DNA (blue), dynein (red), and ASPM-1 (not shown in merge). Cortex is represented by the dashed line. (**D**) Representative images of oocyte spindles (GFP::tubulin and GFP::histone) in germline counting and corresponding quantifications; auxin treatment leads to splayed poles and spindle rotation defects. Cortex is represented by the dashed line. (**E**) IF imaging of Dynein AID conditions showing effects of 4 hr auxin treatment on metaphase (second row) and anaphase (bottom row); shown are tubulin (green), DNA (blue), ASPM-1 (red), and dynein (not shown in merge). ASPM-1 labeling supports initial observations of splayed poles seen in germline counting. (**F**) Quantifications of IF imaging shown in (**E**); meiotic spindles have significantly splayed poles upon auxin treatment. All scale bars = 2.5 µm. Controls for exposure to auxin plates can be found in *Figure 1—figure supplement 1*. Further experimentation with dynein mislocalization through *aspm-1(RNAi)* or *lin-5(RNAi)* can be found in *Figure 1—figure supplement 2*.

The online version of this article includes the following figure supplement(s) for figure 1:

**Figure supplement 1.** Long-term treatment of wild-type worms with auxin does not have adverse effects on oocyte spindle assembly.

**Figure supplement 2.** Oocyte spindles assembled following either ASPM-1 or LIN-5 depletion are morphologically similar to Dynein auxin-inducible degron (AID).

of adult Dynein AID worms on auxin-containing plates, we observed clear defects in centrosome separation and spindle positioning along the A-P axis and dynein was undetectable (22/22 mitotic embryos) (*Figure 1B*), confirming strong depletion in embryos under conditions where germline and worm development are not affected.

To further validate the Dynein AID strain, we assessed the localization of tagged dynein throughout oocyte meiosis and found that the pattern was consistent with previous studies (*Figure 1C*; *Ellefson and McNally, 2009*; *van der Voet et al., 2009*; *Ellefson and McNally, 2011*; *Crowder et al., 2015*). Dynein was weakly associated with forming and bipolar spindles with a slight enrichment at acentrosomal poles, labeled by the microtubule minus end marker ASPM-1, and then became strongly enriched at poles during early anaphase; this enrichment decreased by late anaphase, with a population that was retained near the cortex. Live imaging corroborated our fixed imaging (*Video 1*, n = 3) and also made clearer a population of dynein at kinetochores, as has been shown in other live imaging studies (*McNally et al., 2016*; *Danlasky et al., 2020*).

Given this initial validation, we created a Dynein AID strain that also expressed GFP::tubulin and GFP::histone and then assessed the effects of dynein depletion in oocytes by quantifying spindle morphology in intact worms (*Figure 1D*, *Table 1*). This 'germline counting' method takes advantage of the fact that the *C. elegans* germline is arranged in a production-line fashion (*Wolff et al., 2022*). Oocytes are fertilized in the –1 position of the germline, and then oocyte spindles assemble as the newly fertilized embryo moves through the spermatheca and into the +1 position (*Wolff et al., 2016*).

Under control conditions, bipolar oocyte spindles with focused poles are prevalent by the time the embryo reaches the +1 position (*Figure 1D*; *Wolff et al., 2022*). However, following 4 hr of auxin treatment, most spindles had splayed poles; a majority of spindles displayed microtubule bundles that did not come together at a single focused point on either side of the chromosomes (84/121, 69.4%), similar to what was observed following depletion of dynactin component DNC-1 in a previous study (*Crowder et al., 2015*). Imaging of ASPM-1 confirmed this phenotype at a percentage similar to that observed using germline counting (39/54, 72.2%); following a 4 hr auxin treatment, microtubule minus ends lacked coalesced points between the ends of microtubule bundles (*Figure 1E and F*, *Table 2*). Moreover, we also saw defects in spindle rotation in anaphase (*Figure 1D and E*), consistent with previous studies (*Ellefson and McNally, 2009*; *van der Voet et al., 2009*; *Ellefson and McNally,*

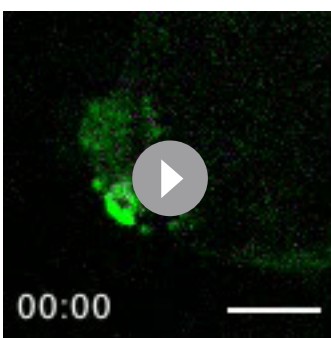

**Video 1.** Localization of dynein during oocyte meiosis. Shows an oocyte expressing mCherry::tubulin, dissected into a control Meiosis Medium solution. DHC-1::degron::GFP localization can be seen at spindle poles during metaphase, alongside faint localization to kinetochores. Localization is consistent across all oocytes filmed (n = 3). Time elapsed shown in min:s. Scale bar = 5 µm.
https://elifesciences.org/articles/72872/figures#video1

**Table 1.** Exact numbers and percentages from germline counting experiment, to assess effects of dynein depletion on oocyte spindle assembly (corresponds to *Figure 1D*).

| Location | Treatment | MT cage | Unfocused | Focused | Anaphase |
|---|---|---|---|---|---|
| −1 position | No auxin | 60/70 (85.7%) | 10/70 (14.3%) | N/A | N/A |
| | With auxin | 47/54 (87.0%) | 7/54 (13.0%) | N/A | N/A |
| Spermatheca | No auxin | N/A | 54/60 (90.0%) | 6/60 (10.0%) | N/A |
| | With auxin | N/A | 63/66 (95.5%) | 3/66 (4.5%) | N/A |
| + 1 position | No auxin | N/A | 8/160 (5.0%) | 97/160 (60.6%) | 55/160 (34.4%) |
| | With auxin | N/A | 84/121 (69.4%) | 8/121 (6.6%) | 29/121 (24.0%) |

*2011*; *Crowder et al., 2015*). To confirm that these defects were not caused by nonspecific effects of auxin, we performed germline counting in a 'TIR1 control' strain that contained all of the same transgenes but lacked degron-tagged DHC-1. In this strain, there was no significant difference between control and auxin-treated worms at any position in the germline, and all stages of spindle assembly appeared identical (*Figure 1—figure supplement 1*, *Table 3*). These findings confirm the prediction that dynein is required for pole focusing during spindle assembly in oocytes and validate the use of the Dynein AID strain to study dynein in oocytes and embryos.

## Delocalization of dynein from spindle poles has similar effects as Dynein AID depletion

The fact that dynein normally localizes to acentrosomal poles and depletion caused pole splaying suggests that dynein's localization is critical to its function in spindle pole formation. To test this hypothesis, we sought to delocalize dynein from poles by depleting two known dynein regulators, ASPM-1 and LIN-5 (NuMA). The MAP NuMA has been shown to direct cytoplasmic dynein to microtubule minus ends to promote pole focusing during mitosis (*Elting et al., 2014*; *Hueschen et al., 2017*), and the *C. elegans* homolog of NuMA, LIN-5, has been shown to target dynein to spindle poles in oocytes to promote spindle rotation (*van der Voet et al., 2009*). Additionally, ASPM-1 is required to properly target LIN-5 to acentrosomal poles, placing it upstream of both LIN-5 and DHC-1 (*van der Voet et al., 2009*). Therefore, we depleted either LIN-5 or ASPM-1 and utilized germline counting to compare oocyte spindle morphologies to Dynein AID (*Figure 1—figure supplement 2A*, *Table 4*). The frequency of pole splaying was nearly identical between these two RNAi conditions and long-term Dynein AID (167/261, 64%, and 364/521, 69.9%, for the +1 position in *lin-5* and *aspm-1* RNAi, respectively, compared to 84/121, 69.4% for Dynein AID); this quantification was in agreement with previous observations of spindle morphology following *aspm-1(RNAi)* or *lin-5(RNAi)* (*van der Voet et al., 2009*; *Wignall and Villeneuve, 2009*; *Connolly et al., 2014*; *Laband et al., 2017*).

To determine if the pole splaying observed following LIN-5 and ASPM-1 depletion was mostly a consequence of dynein mislocalization, we combined these RNAi conditions with Dynein AID depletion (*Figure 1—figure supplement 2B*). Treatment with auxin to remove dynein did not exacerbate the pole defects of either *aspm-1(RNAi)* or *lin-5(RNAi),* nor present any new phenotypes that were not present in the RNAi conditions alone, except for removing the kinetochore population of dynein (pole defects were consistent in 37/37 *lin-5(RNAi)* and 56/56 *aspm-1(RNAi)* oocyte spindles treated with auxin). These results confirm that dynein's localization to poles is necessary for its role in pole coalescence during spindle assembly.

**Table 2.** Exact numbers and percentages from quantification of IF images, to validate the phenotypes observed in germline counting (corresponds to *Figure 1F*).

| Treatment | Focused poles | Unfocused poles | Anaphases |
|---|---|---|---|
| No Auxin | 35/60 (58.3%) | 3/60 (5.0%) | 22/60 (36.7%) |
| With Auxin | 3/54 (5.6%) | 39/54 (72.2%) | 12/54 (22.2%) |

**Table 3.** Exact numbers and percentages from germline counting experiment, to assess effects of auxin treatment on oocyte spindle assembly in a strain that does not contain degron-tagged dynein (corresponds to *Figure 1—figure supplement 1*).

| Location | Treatment | MT cage | Unfocused | Focused | Anaphase |
|---|---|---|---|---|---|
| −1 position | No auxin | 135/146 (92.5%) | 11/146 (7.5%) | N/A | N/A |
| | With auxin | 137/152 (90.1%) | 15/152 (9.9%) | N/A | N/A |
| Spermatheca | No auxin | N/A | 204/222 (91.9%) | 18/222 (8.1%) | N/A |
| | With auxin | N/A | 226/245 (92.2%) | 19/245 (7.8%) | N/A |
| + 1 position | No auxin | N/A | 22/452 (4.9%) | 284/452 (62.8%) | 146/452 (32.3%) |
| | With auxin | N/A | 26/435 (6.0%) | 271/435 (62.3%) | 138/435 (31.7%) |

## Dynein is required throughout meiosis to maintain focused poles

After corroborating that dynein was required to focus acentrosomal poles during spindle assembly, we next sought to determine whether dynein was required to maintain focused poles by performing more rapid depletion ('short-term AID,' *Figure 1A*). We therefore arrested oocytes in metaphase I to enrich for pre-formed spindles (by depleting anaphase-promoting complex component EMB-30; *Furuta et al., 2000*), soaked worms in auxin for 25–30 min to deplete dynein, and then performed IF. Under these conditions, nearly all spindles had severely splayed poles (*Figure 2A and B*); we measured the widths of the spindle equator and the poles and found an approximately 50% increase in the pole to equator ratio upon auxin treatment. Despite this pole splaying, microtubule bundles retained lateral associations with the chromosomes, resulting in a spindle that was still bipolar but more rectangular than wild-type spindles. Notably, we observed these same phenotypes in the absence of *emb-30(RNAi)*, demonstrating that they are not dependent on the metaphase arrest (*Figure 2A and B*).

During this analysis, we also noticed that while ASPM-1 remained localized to dynein-depleted poles it appeared more dispersed throughout the spindle; this suggests a decrease in poleward transport of microtubule minus ends, as has been reported in dynein-depleted mitotic spindles (*Hueschen et al., 2017*). To quantify this phenotype, we performed linescans to measure ASPM-1 levels across the spindle. While ASPM-1 localizes to two major peaks that define the poles of wild-type spindles, it was more diffusely localized across the entire length of the spindle following dynein depletion (*Figure 2—figure supplement 1*). Moreover, we also measured the pole-to-pole lengths of these spindles and found that both unarrested and metaphase-arrested spindles were significantly longer (~3.5 µm) after short-term Dynein AID depletion (*Figure 2—figure supplement 1*). These data suggest that dynein not only promotes focusing of microtubule minus ends into discrete poles, but also provides an inward force that maintains spindle length.

To confirm these results and to assess the dynamics of pole splaying, we coupled rapid dynein depletion with live imaging by dissecting metaphase-arrested oocytes directly into auxin solution and then immediately filming ('acute AID'; *Figure 1A*). We generated a Dynein AID strain expressing mCherry::tubulin so that we could visualize the spindle while dynein, tagged with GFP, was being

**Table 4.** Exact numbers and percentages from germline counting experiment, to compare spindle morphologies of *lin-5/aspm-1(RNAi)* to dynein AID experiments (corresponds to *Figure 1—figure supplement 2A*).

| Location | Condition | Focused poles | Unfocused poles | Anaphases |
|---|---|---|---|---|
| Spermatheca | *control (RNAi)* | 7/103 (6.8%) | 96/103 (93.2%) | N/A |
| | *lin-5 (RNAi)* | 7/135 (5.2%) | 128/135 (94.8%) | N/A |
| | *aspm-1 (RNAi)* | 9/202 (4.5%) | 193/202 (95.5%) | N/A |
| + 1 position | *control (RNAi)* | 124/191 (64.9%) | 9/191 (4.7%) | 58/191 (30.4%) |
| | *lin-5 (RNAi)* | 12/261 (4.6%) | 167/261 (64.0%) | 82/261 (31.4%) |
| | *aspm-1(RNAi)* | 18/521 (3.5%) | 364/521 (69.9%) | 139/521 (26.6%) |

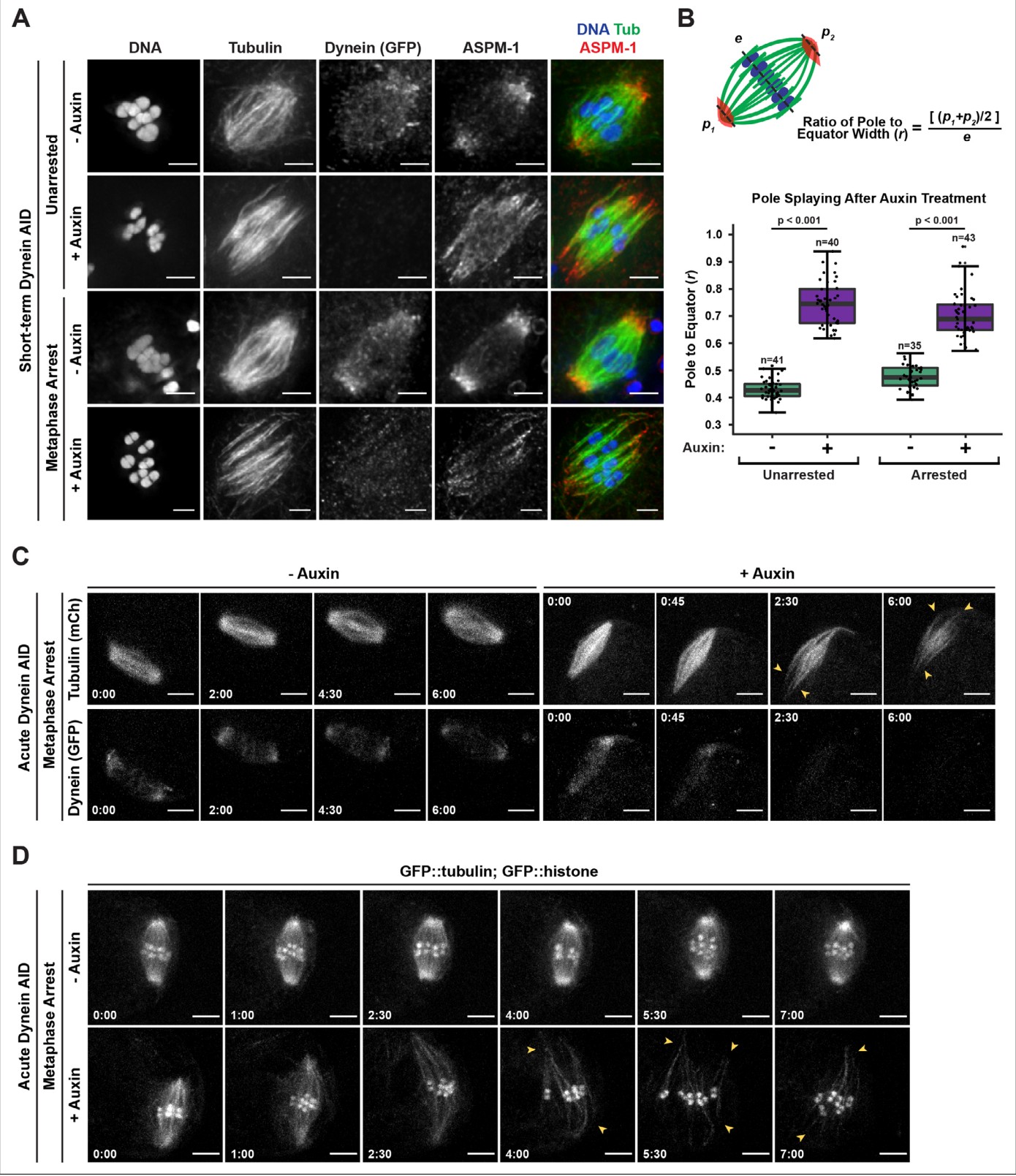

**Figure 2.** Dynein is required to maintain focused spindle poles. (**A**) Immunofluorescence (IF) imaging of oocyte spindles in control or metaphase I-arrest (*emb-30(RNAi)*) conditions; shown are tubulin (green), DNA (blue), ASPM-1 (red), and dynein (not shown in merge). Acentrosomal poles become unfocused upon acute dynein depletion. Scale bars = 2.5 µm. (**B**) Quantification of the ratio between the width of the spindle equator and the width of the spindle poles shows that addition of auxin causes nearly all spindle poles to unfocus and splay, both with and without the metaphase arrest.

*Figure 2 continued on next page*

*Figure 2 continued*

Statistical significance between means of pole splaying was determined via a two-tailed *t*-test for unarrested and arrested conditions, with a p-value of $<2.2 \times 10^{-16}$ for both. Further quantification of changes in spindle length and microtubule minus end distribution upon auxin treatment can be seen in *Figure 2—figure supplement 1A and B*. (**C**) Ex utero live imaging of metaphase-arrested spindles; vehicle-treated control oocytes are shown on the left, auxin-treated oocytes are shown on the right. Addition of auxin causes rapid depletion of dynein (labeled with GFP, bottom row) and dynamic splaying of both spindle poles (shown using mCherry::tubulin, top row; arrowheads). Time elapsed shown in min:s. Scale bars = 5 μm. (**D**) Ex utero live imaging of metaphase-arrested spindles, shown using GFP::tubulin and GFP::histone; addition of auxin (bottom row) causes dynamic splaying of poles (arrowheads) but does not grossly disrupt chromosome alignment; vehicle-treated control oocytes are shown on the top row. Time elapsed shown in min:s. Scale bars = 5 μm. Controls for effects of auxin alone in ex utero imaging can be seen in *Figure 2—figure supplement 2*. Additional ex utero imaging of unarrested oocytes treated with auxin can be seen in *Figure 2—figure supplement 3*.

The online version of this article includes the following source data and figure supplement(s) for figure 2:

**Source data 1.** The source data for *Figure 2B* is provided.

**Figure supplement 1.** Acute dynein depletion leads to dispersion of ASPM-1 across the spindle and increased pole-to-pole spindle length.

**Figure supplement 1—source data 1.** The spindle length measurements for unarrested (*control(RNAi)*) and metaphase I-arrested (*emb-30(RNAi)*) spindles are listed in separate tabs.

**Figure supplement 1—source data 2.** The measurements of ASPM-1 intensity across each spindle for unarrested (*control(RNAi)*) and metaphase I-arrested (*emb-30(RNAi)*) spindles are listed in separate tabs.

**Figure supplement 2.** Treatment of wild-type oocytes with auxin during ex utero imaging does not perturb spindle architecture or organization.

**Figure supplement 2—source data 1.** The measurements for spindle length and pole/equator width are listed in separate tabs.

**Figure supplement 3.** Phenotypes seen in dynein depletion are not an artifact of metaphase arrest.

depleted. In the absence of auxin, there was no noticeable change in dynein localization or brightness throughout the imaging time frame (*Figure 2C*, *Video 2*, n = 4). In contrast, when oocytes were dissected into auxin, dynein appeared depleted from meiotic spindles within roughly 3 min and acentrosomal poles began to splay (*Figure 2C*, *Video 3*, n = 4). Splaying occurred at both poles (*Figure 2C*, arrowheads) and spindle length increased over time. To visualize the movement and alignment of chromosomes during this process, we repeated this experiment in our GFP::tubulin and GFP::histone Dynein AID strain (*Figure 2D*, *Video 4*, n = 7). As before, metaphase-arrested spindles were quickly disrupted upon dissection into auxin solution (*Video 5*, n = 8); poles began to splay (*Figure 2D*, arrowheads) and spindles again lengthened. Despite these changes, microtubule bundles were able to remain associated with chromosomes and the spindles maintained bipolarity.

To ensure that these phenotypes were not caused by auxin itself having nonspecific effects on dissected oocytes, we performed analogous live imaging experiments in our TIR1 control strain (lacking degron-tagged DHC-1) to assess the effects of auxin treatment in the absence of dynein depletion. Neither untreated nor auxin-treated oocytes exhibited significant changes to either spindle length or pole to equator ratio (n = 5 for each condition); we compared both individual traces and average change from baseline for both of these parameters (*Figure 2—figure supplement 2*, *Videos 6 and 7*). Additionally, we repeated these experiments without *emb-30(RNAi)* and saw the same pole defects in the absence of the metaphase arrest (*Figure 2—figure supplement 3*, *Videos 8–11*). These auxin-treated oocytes progressed through anaphase despite pole splaying, albeit with lagging chromosomes, as has been observed previously following dynein depletion (*Muscat et al., 2015*; *McNally et al., 2016*). Moreover, there were defects in spindle rotation (n = 9), consistent with prior studies (*Ellefson and McNally, 2009*; *van der Voet et al., 2009*; *Ellefson and McNally, 2011*; *Crowder et al., 2015*). Altogether, these data demonstrate that dynein is not only required to focus poles during meiotic spindle assembly, but is also required to maintain the integrity of acentrosomal poles once they have been established.

## Removal of dynein from monopolar spindles causes catastrophic breakdown of spindle architecture

Despite the pole splaying observed following dynein depletion, spindle microtubules remained aligned along the same axis and bipolarity was not lost. Therefore, we hypothesized that microtubule bundles in the central region imparted stability to the entire spindle. To test this directly, we sought to deplete dynein from a monopolar spindle, generated by depleting the kinesin-12 motor KLP-18;

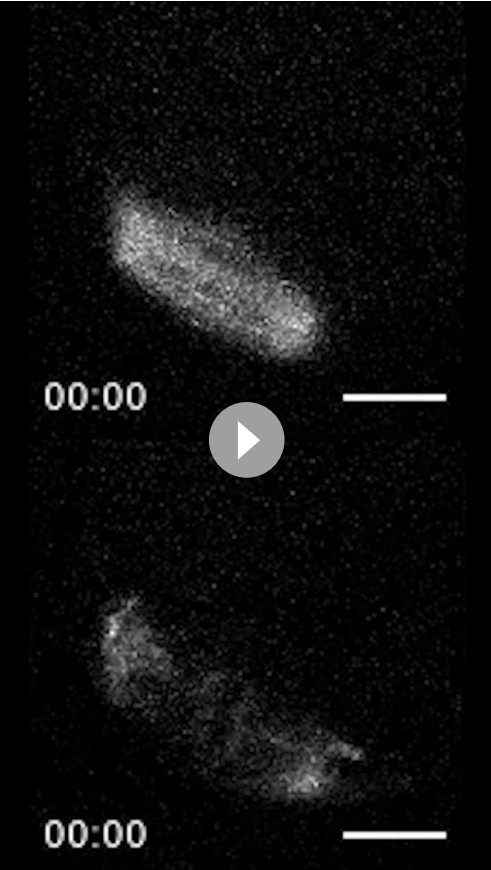

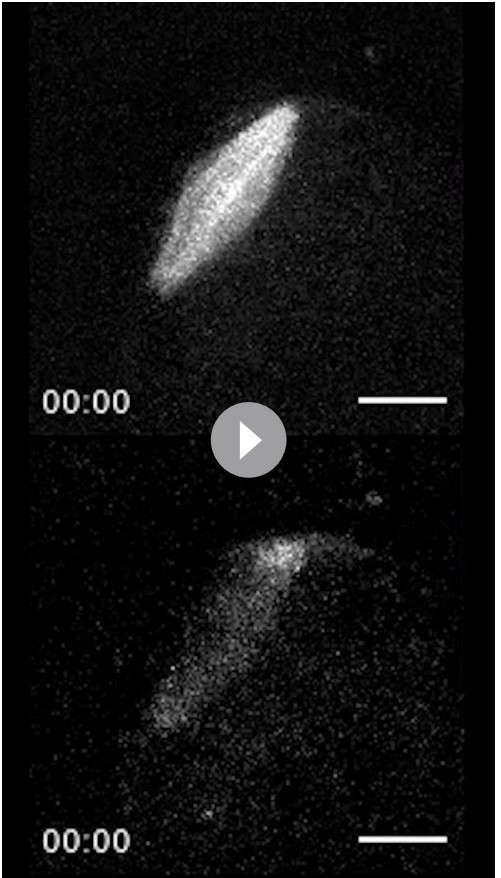

**Video 2.** Dynein localizes to spindle poles in the Dynein AID strain in the absence of auxin. Shows a metaphase-arrested *(emb-30(RNAi))* oocyte expressing mCherry::tubulin (top), dissected into control Meiosis Medium solution. DHC-1::degron::GFP (bottom) is clearly localized to spindle poles and faintly seen at kinetochores and spindle morphology does not change during the imaging timeframe in the absence of auxin. Corresponds to Figure 2C. Results are consistent across all oocytes filmed (n = 4). Time elapsed shown in min:s. Scale bar = 5 µm.

https://elifesciences.org/articles/72872/figures#video2

**Video 3.** Auxin treatment of Dynein AID worms rapidly depletes dynein and unfocuses poles. Shows a metaphase-arrested *(emb-30(RNAi))* oocyte expressing mCherry::tubulin (top), dissected into Meiosis Medium containing 100 µM auxin to deplete dynein. Once dissected into auxin solution, rapid depletion of DHC-1::degron::GFP (bottom) is evident within 3 min, at which point spindle poles begin to unfocus and splay apart. Corresponds to Figure 2C. Phenotypes are consistent across all oocytes filmed (n = 4). Time elapsed shown in min:s. Scale bar = 5 µm.

https://elifesciences.org/articles/72872/figures#video3

these structures have a single focused pole and therefore no area of microtubule overlap that could theoretically stabilize the structure if the single pole is disrupted (*Wignall and Villeneuve, 2009*). Previously, we found that partial dynein RNAi on a monopolar spindle led to chromosomes moving a substantial distance away from the center of intact monopoles (*Muscat et al., 2015*); the AID approach now allows us to revisit these results without limiting the amount of dynein depletion to avoid developmental defects.

First, we performed long-term Dynein AID in worms treated with *klp-18(RNAi)* and found that oocytes from these worms rarely displayed a monopolar spindle (*Figure 3A*, *Table 5*). Instead, oocyte chromosomes were dispersed, retaining lateral associations to microtubule bundles (116/125 in the +1 position, 92.8%). IF imaging following short-term AID treatment confirmed this result; ASPM-1 labeling revealed that monopoles had begun breaking apart, ejecting chromosomes and associated microtubule bundles into the cytoplasm (*Figure 3B*). In some cases, the entirety of the monopole disappeared, and each chromosome retained its own associated microtubule bundles. Many of these structures had a distinct cone shape, with microtubule ends focused on one side of the chromosome and splayed on the other side (*Figure 3B*, zooms); we speculate that the microtubule minus ends are coalesced together at the pointy

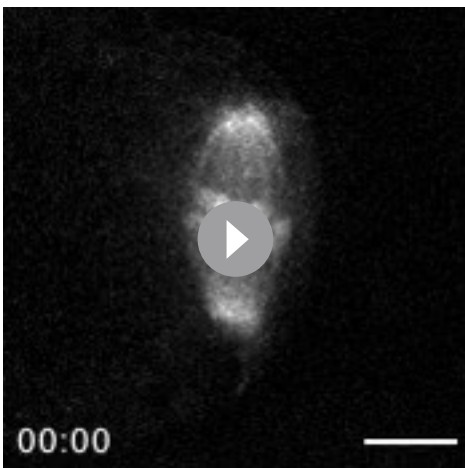

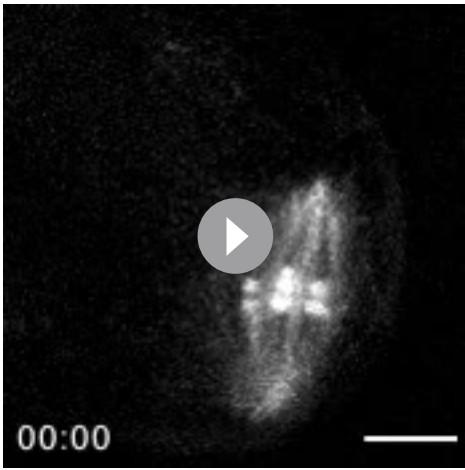

**Video 4.** Metaphase-arrested spindles in the Dynein AID strain remain stable and maintain chromosome alignment in the absence of auxin. Shows a metaphase-arrested oocyte expressing GFP::tubulin and GFP::histone in the Dynein AID strain (where DHC-1 is degron-tagged and TIR1 is expressed in the germ line), dissected into control Meiosis Medium solution. No major changes in spindle length or shape occur, and chromosomes are stably aligned in the spindle center. Corresponds to Figure 2D. Phenotypes are consistent across all oocytes filmed (n = 7). Time elapsed shown in min:s. Scale bar = 5 μm. https://elifesciences.org/articles/72872/figures#video4

**Video 5.** Auxin treatment unfocuses poles, but microtubules largely remain aligned along a single axis. Shows a metaphase-arrested oocyte expressing GFP::tubulin and GFP::histone in the Dynein AID strain (where DHC-1 is degron-tagged and TIR1 is expressed in the germ line), dissected into Meiosis Medium containing 100 μM auxin to deplete dynein. Upon dissection into auxin solution, rapid and dynamic splaying of acentrosomal poles occurs alongside a notable increase in spindle length, but microtubule bundles and chromosomes stay associated and aligned along a single axis. Corresponds to Figure 2D. Phenotypes are consistent across all oocytes filmed (n = 8). Time elapsed shown in min:s. Scale bar = 5 μm. https://elifesciences.org/articles/72872/figures#video5

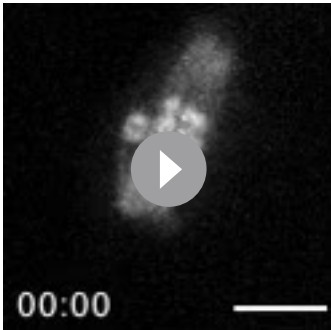

**Video 6.** Metaphase-arrested spindles in the TIR1 control strain are stable and maintain chromosome alignment in the absence of auxin. Shows a metaphase-arrested oocyte expressing GFP::tubulin and GFP::histone in the TIR1 control strain (that contains TIR1 but lacks a degron tag on DHC-1). Dissection into control Meiosis Medium yields no significant change in spindle length or organization. Notably, the spindle appears identical to those observed in control oocytes containing GFP-tagged DHC-1. Corresponds to Figure 2—figure supplement 2A. Phenotypes are consistent across all oocytes filmed (n = 5). Time elapsed shown in min:s. Scale bar = 5 μm. https://elifesciences.org/articles/72872/figures#video6

end of this structure, retaining some association even after they were released from the monopole upon spindle breakdown.

We observed the same phenotype using acute dynein depletion coupled with live ex utero imaging; when oocytes containing monopolar spindles were dissected into auxin, we observed rapid breakdown of the monopole on a timescale similar to that required for dynein depletion from bipolar spindles (*Figure 4A and B*, *Video 12* and *Video 13*, n = 5). As the monopolar spindle began to fragment, microtubule bundles remained associated with individual chromosomes via lateral associations (*Figure 4A*, arrowheads), as seen in fixed imaging (*Figure 3B*). These results demonstrate an essential role for dynein in maintaining pole integrity and suggest that in the case of dynein depletion from bipolar spindles the intact overlap zone stabilizes the spindle even though poles are completely disrupted.

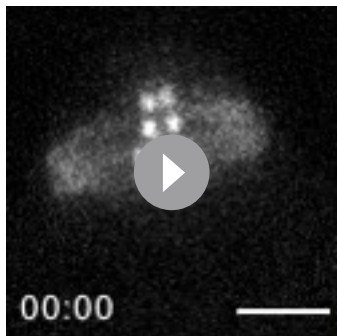

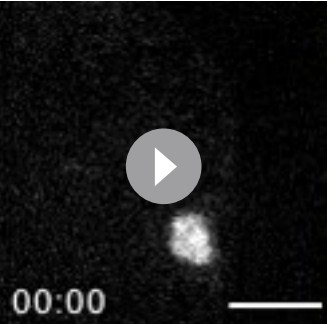

**Video 7.** Addition of auxin to metaphase-arrested oocytes in the TIR1 control strain does not perturb spindle architecture.
Shows a metaphase-arrested oocyte expressing GFP::tubulin and GFP::histone in the TIR1 control strain (that contains TIR1 but lacks a degron tag on DHC-1). Dissection into Meiosis Medium containing 100 µM auxin yields no significant change in spindle length or organization, demonstrating that auxin addition itself is not responsible for phenotypes observed in our acute Dynein auxin-inducible degron (AID) imaging. Corresponds to Figure 2—figure supplement 2A. Phenotypes are consistent across all oocytes filmed (n = 5). Time elapsed shown in min:s. Scale bar = 5 µm.
https://elifesciences.org/articles/72872/figures#video7

**Video 8.** Normal oocytes undergo two rounds of bipolar spindle formation and anaphase segregation. Shows an unarrested oocyte expressing GFP::tubulin and GFP::histone, dissected into control Meiosis Medium solution. After one round of segregation, a second bipolar spindle is formed, rotates perpendicular to the cortex, and successfully extrudes a second polar body. Corresponds to the movie in row 1 of Figure 2—figure supplement 3. Phenotypes are consistent across all oocytes filmed (n = 8). t = 0 is set to onset of meiosis II. Time elapsed shown in min:s. Scale bar = 5µm.
https://elifesciences.org/articles/72872/figures#video8

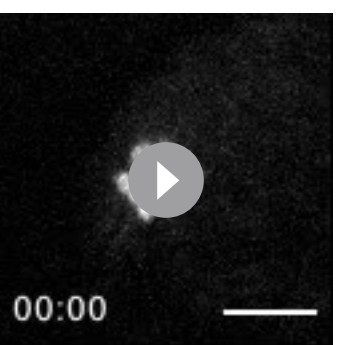

**Video 9.** Normal oocytes undergo two rounds of bipolar spindle formation and anaphase segregation. Another example of an unarrested oocyte expressing GFP::tubulin and GFP::histone, dissected into control Meiosis Medium solution. After one round of segregation, a second bipolar spindle is formed, rotates perpendicular to the cortex, and successfully extrudes a second polar body. Corresponds to the movie in row 2 of Figure 2—figure supplement 3. Phenotypes are consistent across all oocytes filmed (n = 8). t = 0 is set to onset of meiosis II. Time elapsed shown in (min):(sec). Scale bar = 5 µm.
https://elifesciences.org/articles/72872/figures#video9

**Video 10.** Splaying of acentrosomal poles occurs in unarrested oocytes depleted of dynein. Shows an unarrested oocyte expressing GFP::tubulin and GFP::histone, dissected into Meiosis Medium containing 100µM auxin to deplete dynein. A spindle tries to form in the absence of dynein, leading to dynamic and unfocused poles. Anaphase segregation still occurs, albeit with a failure in spindle rotation. Corresponds to the movie in row 3 of Figure 2—figure supplement 3. Phenotypes are consistent across all oocytes filmed (n = 9). t = 0 is set to onset of meiosis II. Time elapsed shown in min:s. Scale bar = 5µm.
https://elifesciences.org/articles/72872/figures#video10

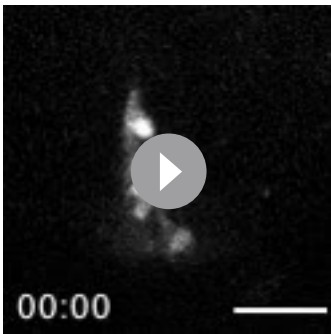

00:00

**Video 11.** Splaying of acentrosomal poles occurs in unarrested oocytes depleted of dynein. Another example of an unarrested oocyte expressing GFP::tubulin and GFP::histone, dissected into Meiosis Medium containing 100 µM auxin to deplete dynein. A spindle tries to form in the absence of dynein, leading to dynamic and unfocused poles. Anaphase segregation still occurs, albeit with a failure in spindle rotation. Corresponds to the movie in row 4 of *Figure 2—figure supplement 3*. Phenotypes are consistent across all oocytes filmed (n = 9). t = 0 is set to onset of meiosis II. Time elapsed shown in (min):(sec). Scale bar = 5 µm

https://elifesciences.org/articles/72872/figures#video11

## Individualized chromosomes can undergo an anaphase-like segregation in the absence of KLP-18 and dynein

In addition to demonstrating the dynamics of monopole breakdown, our live imaging of acute dynein depletion from monopolar spindles revealed an intriguing phenotype. These experiments were not done using metaphase arrest, so in the absence of auxin, chromosomes extended outward from the monopole towards microtubule plus ends, and then moved back towards the monopole during anaphase (*Figure 4A*, *Video 14*, n = 6), as has been described previously (*Muscat et al., 2015*). In auxin-treated oocytes, as filming continued past the full breakdown of the monopole, we were surprised to see that each individual chromosome began to segregate (*Figure 4A and B*, *Videos 12 and 13*, n = 5). This segregation was incredibly striking as it occurred synchronously across the oocyte and each set of separating chromosomes exhibited distinct bidirectional movement, reminiscent of normal chromosome segregation. The fact that we observed these anaphase-like movements in the absence of both KLP-18 and dynein suggests that neither motor is absolutely required for chromosome segregation in oocytes.

To assess whether these segregations were similar to normal anaphases, we first quantified the segregation distances of individual chromosomes and compared them to both wild-type and dynein-depleted anaphases (*Figure 4C*). Chromosomes on dynein-depleted *klp-18(RNAi)* mini anaphase spindles segregated to a similar final distance as wild-type spindles (~5 µm for both wild-type and mini anaphases). However, mini anaphases did not appear to segregate as far as dynein-depleted bipolar anaphases, where KLP-18 was not depleted (~6.5 µm). This observation raises the possibility that, even though chromosomes can segregate without KLP-18, this motor may normally contribute to anaphase spindle elongation through microtubule sliding. This contribution has been difficult to test in previous studies due to the monopolar spindle phenotype that is quickly generated upon removal of KLP-18 (*Wolff et al., 2021*), which prevents normal anaphase segregation from occurring.

To further characterize these segregations, we performed fixed imaging to generate higher-resolution images. This analysis revealed that the mini anaphases have key hallmarks of normal anaphase spindles; when chromosomes were still close together, we observed lateral microtubule bundles running alongside the separating chromosomes (*Figure 5*, 'early anaphase'), and as chromosomes segregated further and spindle length increased, microtubules were largely localized between chromosomes (*Figure 5*, 'late anaphase'). This suggests that, like normal anaphase spindles, these mini spindles undergo morphological changes and anaphase-B-like spindle elongation as they drive chromosomes apart.

Additionally, we used fixed imaging to assess the localization of other proteins that are present during normal anaphase. First, we imaged the ring complex (RC), a collection of proteins that forms a ring around the center of each chromosome in oocytes. RCs aid in chromosome congression (*Wignall and Villeneuve, 2009*; *Muscat et al., 2015*; *Pelisch et al., 2017*; *Hollis et al., 2020*) and then are released from chromosomes during anaphase; they remain as a ring between segregating chromosomes and then are disassembled as anaphase progresses (*Dumont et al., 2010*; *Muscat et al., 2015*; *Davis-Roca et al., 2017*; *Pelisch et al., 2017*; *Davis-Roca et al., 2018*; *Pelisch et al., 2019*). To assess RC behavior, we imaged SUMO, a post-translational modification found in the RC (*Pelisch et al., 2017*). In control and dynein depletion conditions, SUMO-marked RCs were removed from chromosomes during anaphase and remained between segregating chromosomes, as expected

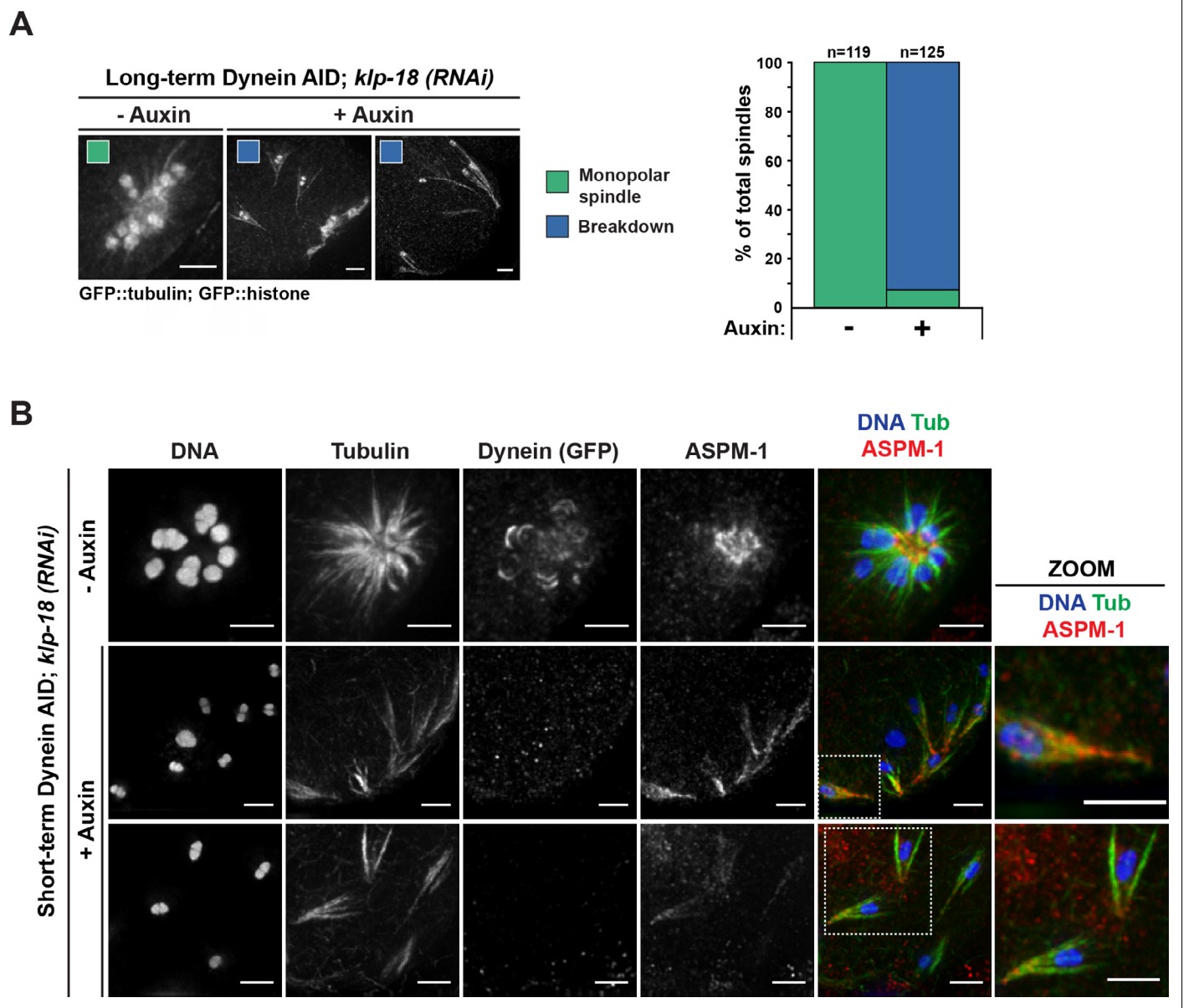

**Figure 3.** Monopolar spindles break down upon dynein depletion. (**A**) Representative images of *klp-18(RNAi)* oocyte spindles (GFP::tubulin and GFP::histone) in germline counting and corresponding quantifications; dynein depletion leads to dissolution of monopoles, releasing chromosomes with associated microtubule bundles into the cytoplasm. Scale bars = 2.5 µm (**B**) Immunofluorescence (IF) imaging of monopole breakdown after dynein depletion; shown are tubulin (green), DNA (blue), ASPM-1 (red), and dynein (not shown in merge). Chromosomes released into the cytoplasm retain lateral microtubule associations (zooms). Scale bars = 2.5 µm; note that the +Auxin images are zoomed out more since chromosomes are more dispersed following monopole breakdown and a larger region needed to be imaged.

**Table 5.** Exact numbers and percentages from germline counting experiment, to assess effects of dynein depletion on monopolar spindles (corresponds to *Figure 3A*).

| Condition | Treatment | Monopolar | Breakdown |
|---|---|---|---|
| *klp-18 (RNAi)* | No auxin | 119/119 (100.0%) | 0/119 (0.0%) |
| | With auxin | 9/125 (7.2%) | 116/125 (92.8%) |

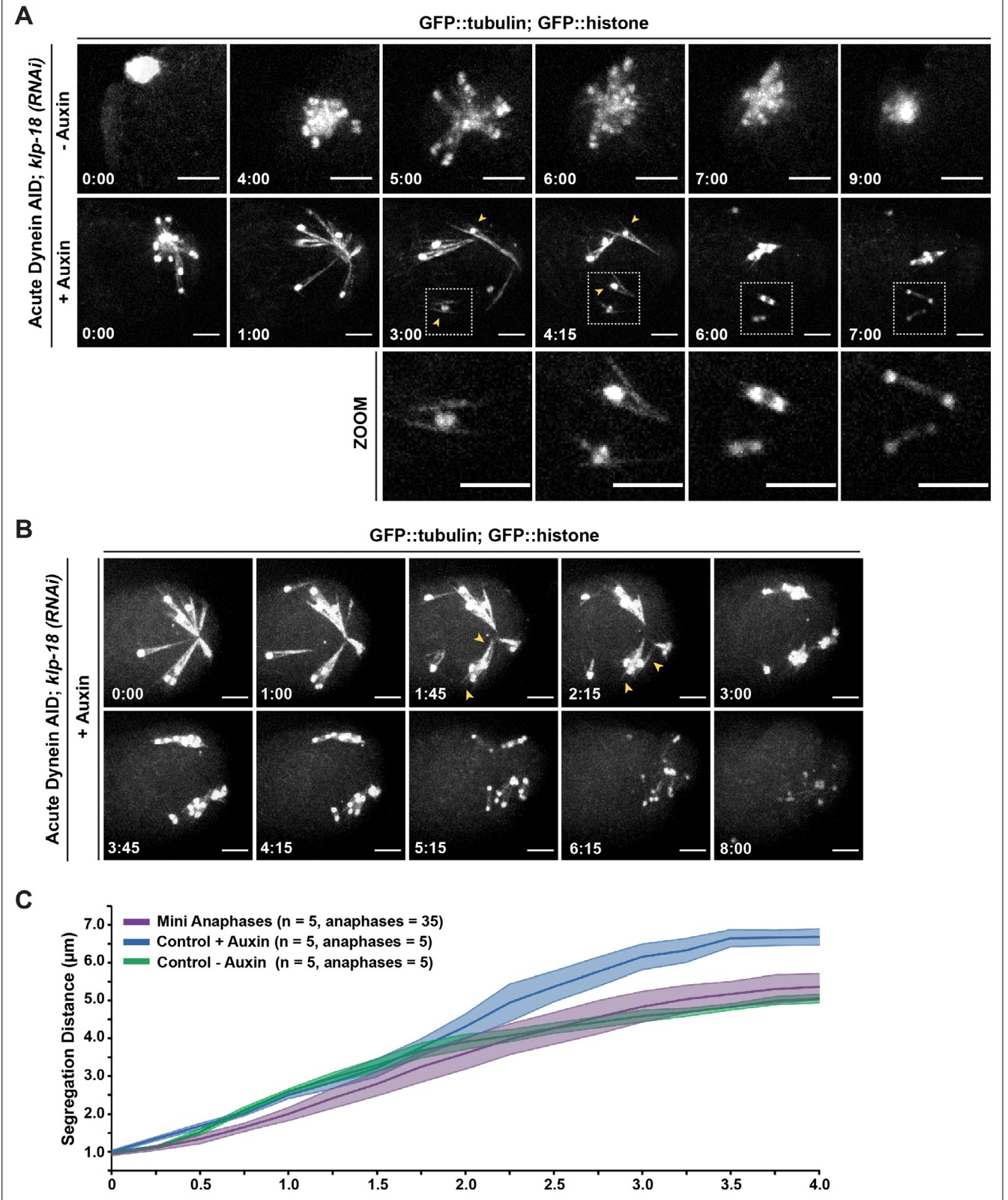

**Figure 4.** Individualized chromosomes undergo anaphase-like segregation in the absence of KLP-18 and dynein. (**A**) Ex utero live imaging of oocytes expressing GFP::tubulin and GFP::histone in control and *klp-18(RNAi)* conditions. In the control (top row), a monopolar spindle forms and then chromosomes move back towards the monopole in anaphase, as previously described (*Muscat et al., 2015*). Addition of auxin to acutely deplete dynein (row 2) leads to rapid breakdown of monopolar spindles. Intriguingly, individualized chromosomes undergo an anaphase-like segregation

*Figure 4 continued on next page*

*Figure 4 continued*

post-breakdown (zooms, row 3). Microtubule bundles remain laterally associated with chromosomes (arrowheads) prior to anaphase-like segregation. Time elapsed shown in min:s. Scale bars = 5 µm. (**B**) Another example of acute auxin treatment, to remove dynein, from an oocyte expressing GFP::tubulin and GFP::histone in *klp-18(RNAi)* conditions; after breakdown of the monopole and reorganization of microtubules (arrowheads), individual chromosomes are able to undergo synchronized anaphase-like segregation. Time elapsed shown in min:s. Scale bars = 5 µm. (**C**) Quantification of timelapses from control anaphases (the Dynein auxin-inducible degron [AID] strain without auxin), anaphases following dynein depletion (the Dynein AID strain with auxin), and miniature anaphases. Each timelapse was synchronized to initiation of anaphase A, and the distance between the center of each chromosome was measured at each 15 s timepoint. For each condition, the number of timelapses used is represented by n, and the shaded area around each average line represents the SEM. Note that for the mini anaphase condition, though we used five movies, we measured multiple mini anaphases in each (total anaphases = 35). Miniature anaphases do not exhibit significant differences in segregation rates or distances to wild-type anaphase spindles, but do not reach distances of dynein depletions alone.

The online version of this article includes the following source data for figure 4:

**Source data 1.** The source data for *Figure 4C* is provided.

(*Figure 5—figure supplement 1*). During monopolar spindle breakdown, SUMO localized to RCs as individual chromosomes detached from the dissolving monopole. In mini anaphases, SUMO marked a singular disassembling RC between each segregating chromosome pair (93/93 oocytes observed), congruent with the normal behavior of RCs in anaphase.

Next, we assessed SEP-1 (separase), a protease that cleaves a component of the cohesin complex, enabling homologous chromosomes (meiosis I) and sister chromatids (meiosis II) to separate (*Siomos et al., 2001*; *Kudo et al., 2006*). During wild-type meiosis, SEP-1 localizes to kinetochores in metaphase and relocalizes to RCs at anaphase onset; by the end of anaphase, SEP-1 is no longer detectable on the anaphase spindle (*Muscat et al., 2015*; *Davis-Roca et al., 2017*). We observed this same pattern when assessing monopolar breakdowns following *klp-18(RNAi)* and short-term Dynein AID.

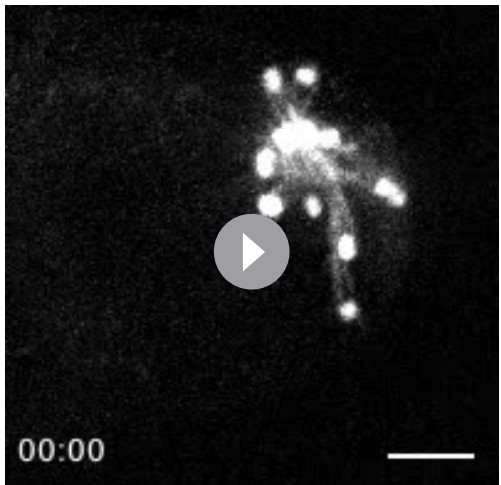

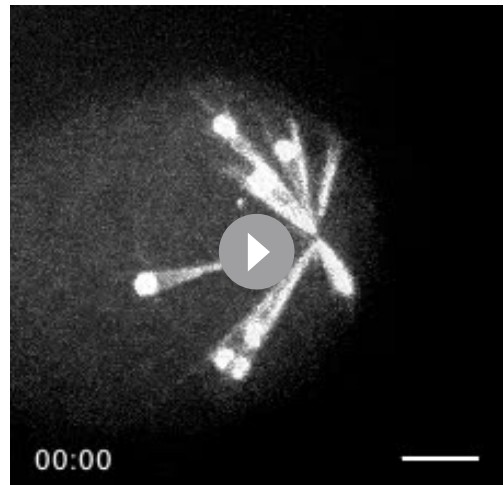

**Video 12.** Monopolar spindles break down and individual chromosomes segregate following dynein depletion. Shown is an oocyte from the Dynein AID strain expressing GFP::tubulin and GFP::histone following *klp-18(RNAi)* to generate monopolar spindles, dissected into Meiosis Medium containing 100 µM auxin to deplete dynein. Upon dissection into auxin solution, the monopole quickly dissolves, leading to ejection of individual chromosomes (with laterally associated microtubule bundles) into the cytoplasm, followed by a synchronous anaphase-like segregation event. Corresponds to Figure 4A. Phenotypes are consistent across all oocytes filmed (n = 5). Time elapsed shown in min:s. Scale bar = 5 µm.
https://elifesciences.org/articles/72872/figures#video12

**Video 13.** Monopolar spindles break down and individual chromosomes segregate following dynein depletion. Another example of a Dynein AID oocyte expressing GFP::tubulin and GFP::histone following *klp-18(RNAi)*, dissected into Meiosis Medium containing 100 µM auxin to deplete dynein. In this example, reorganization of local microtubule bundles around individual chromosomes can be observed after breakdown of the monopolar spindle. Shortly after reorganization of these microtubules, anaphase-like segregation occurs synchronously across all chromosomes. Corresponds to Figure 4B. Phenotypes are consistent across all oocytes filmed (n = 5). Time elapsed shown in min:s. Scale bar = 5 µm.
https://elifesciences.org/articles/72872/figures#video13

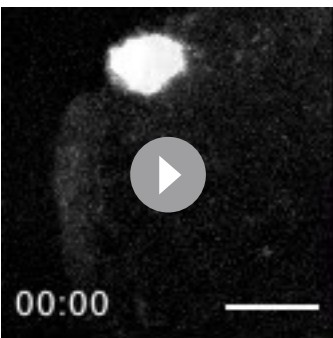

**Video 14.** Dynamics of a normal monopolar spindle. Shows an oocyte expressing GFP::tubulin and GFP::histone following *klp-18(RNAi)*, dissected into a Meiosis Medium control solution. Chromosomes extend outward towards microtubule plus ends forming a monopolar spindle during metaphase, but retract towards the minus ends during anaphase. Corresponds to Figure 4A. Phenotypes are consistent across all oocytes filmed (n = 6). Time elapsed shown in min:s. Scale bar = 5 μm.
https://elifesciences.org/articles/72872/figures#video14

After monopole breakdown, SEP-1 relocalized from kinetochores to RCs, and then was absent from each mini spindle in late anaphase (81/81 oocytes observed) (*Figure 5—figure supplement 2*). Taken together, these data demonstrate that miniature spindles recapitulate the key aspects of normal anaphase progression.

## Microtubules can reorganize into mini bipolar spindles in the absence of dynein and KLP-18

The fact that we observed chromosome segregation with hallmarks of normal anaphase led us to hypothesize that some local microtubule reorganization must be occurring after monopole breakdown; without a bipolar distribution of microtubule minus ends, it should not be possible for chromosomes to move away from each other in the manner we observed. Therefore, while all microtubule minus ends are together in the monopole before dynein depletion, we speculated that these microtubules may reorganize into a bipolar structure around each chromosome following monopole breakdown. In order to better characterize this possible reorganization, we assessed the localization of ASPM-1 as a marker of microtubule minus ends. On a few occasions (n = 3 oocytes), we were able to capture what appeared to be a transitional period, indicating that reorganization was occurring; ASPM-1 could be seen on both sides of these mini spindles, with minor enrichment towards areas resembling spindle poles (*Figure 6A*). Note that while ASPM-1 also appeared to localize on chromosomes in these experiments, this has been previously shown to be nonspecific cross-reactivity of this particular antibody (*Wignall and Villeneuve, 2009*).

This imaging suggested that microtubules might reorganize into a mini bipolar spindle with overlapping antiparallel microtubules in the center, which could facilitate chromosome segregation. To test this idea, we assessed the localization of a well-studied spindle protein, the PRC1 homolog SPD-1. SPD-1 localizes to the anaphase spindle midzone in oocytes, marking a region of antiparallel microtubule overlap (*Hattersley et al., 2016*; *Gigant et al., 2017*; *Mullen et al., 2017*; *Figure 6B*, row 2), and we found that SPD-1 was also at this location in the absence of dynein (*Figure 6B*, row 4). In intact and disassembling monopoles, SPD-1 was not localized to microtubules as it only associates with the spindle in anaphase. However, once chromosomes had been ejected into the cytoplasm, SPD-1 could be clearly observed between segregating chromosomes (*Figure 6B*, bottom row, 122/122 oocytes observed), demonstrating that the mini anaphase spindles have a region of antiparallel microtubule overlap in the center.

## The kinesin-5 family motor BMK-1 provides an outward sorting force that allows spindle reorganization in the absence of KLP-18 and dynein

Our discovery that microtubules were able to reorganize around dispersed chromosomes and establish a region of antiparallel overlap at the center of each mini spindle suggested the presence of an activity that can sort microtubules and re-establish bipolarity in the absence of KLP-18 and dynein. We saw this condition as a unique opportunity to probe for redundant motor forces that may be present in a normal meiotic spindle, but would otherwise be masked by the presence of the forces provided by KLP-18 and dynein. One strong candidate was the sole kinesin-5 family motor in *C. elegans*, BMK-1 (*Bishop et al., 2005*). Kinesin-5 motors have essential roles in establishing spindle bipolarity in many organisms (reviewed in *Mann and Wadsworth, 2019*), but loss of BMK-1 has not been reported to have major phenotypes in *C. elegans*; BMK-1 depletion has no effects on spindle morphology in either oocytes or embryos (*Bishop et al., 2005*) and only minor effects on chromosome segregation

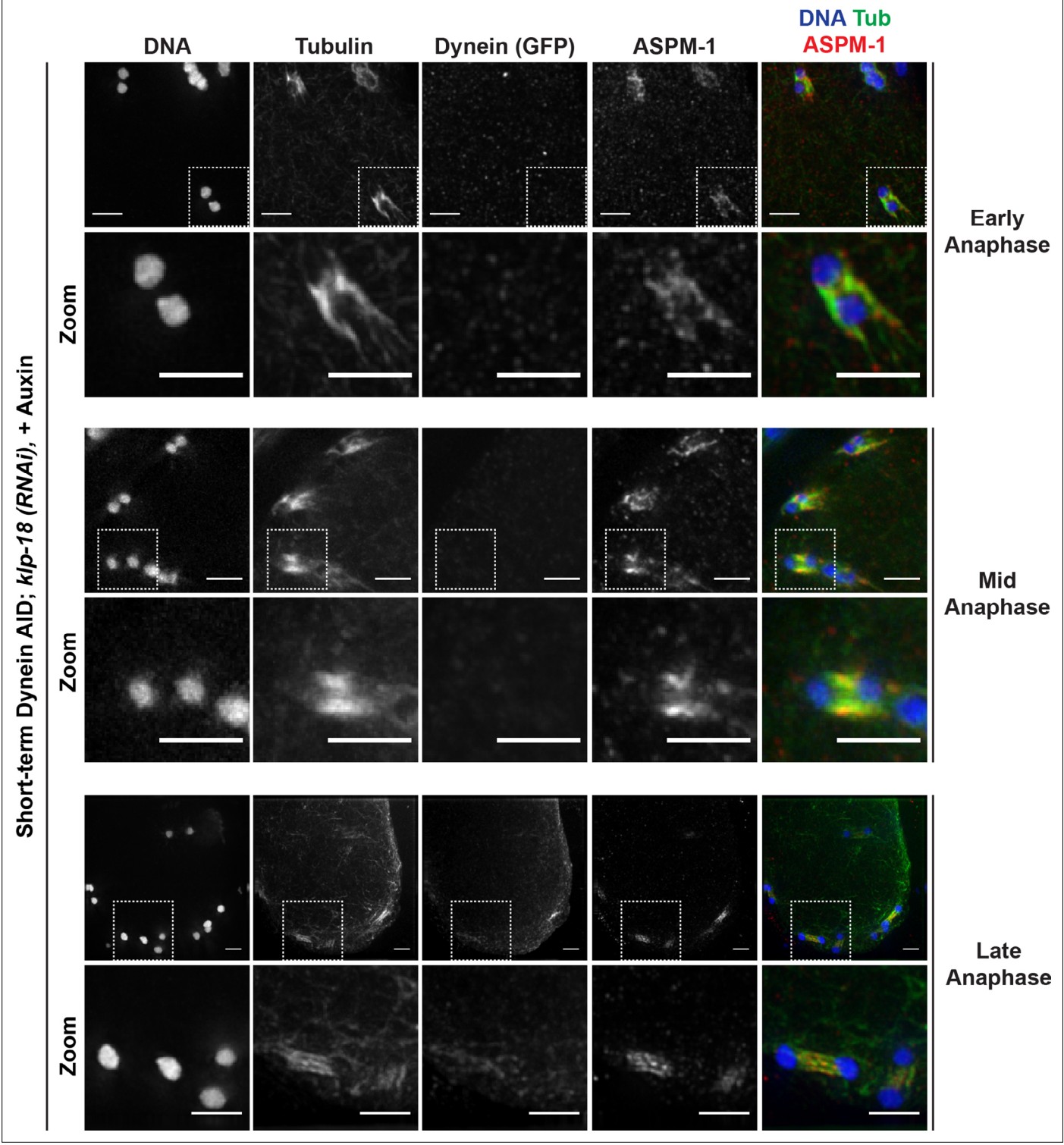

**Figure 5.** Miniature anaphase spindles recapitulate key features of normal anaphase progression. Immunofluorescence (IF) imaging of miniature anaphases at multiple chromosome segregation distances, representing various stages of anaphase; shown are tubulin (green), DNA (blue), ASPM-1 (red), and dynein (not shown in merge). For each stage, a zoomed-out view is shown (top row), as well as a zoom of particular mini anaphase spindles (bottom row). Segregation events closely resemble different stages of anaphase A and anaphase B. Scale bars = 2.5 μm. Further characterization of miniature anaphases via localization of SUMO and separase can be found in *Figure 5—figure supplement 1* and *Figure 5—figure supplement 2*, respectively.

The online version of this article includes the following figure supplement(s) for figure 5:

*Figure 5 continued on next page*

*Figure 5 continued*

**Figure supplement 1.** SUMO is localized normally in all dynein depletion conditions.

**Figure supplement 2.** Separase localizes normally to miniature anaphase spindles.

rates (*Saunders et al., 2007*; *Laband et al., 2017*). We therefore hypothesized that BMK-1 could be providing a supplementary outward sorting force that is normally masked by the contributions of KLP-18.

To probe this hypothesis, we first sought to confirm the localization of BMK-1 on meiotic spindles and to test if this motor was localized to microtubules under monopole breakdown conditions. BMK-1 was broadly associated with bipolar metaphase and anaphase spindles in both control and dynein-depletion conditions (*Figure 7A*). In intact monopolar spindles, BMK-1 also localized to microtubules. Importantly, when we depleted dynein and induced monopole breakdown, BMK-1 still localized to microtubules (*Figure 7A*, zooms), placing it in a location where it could contribute to microtubule reorganization (150/150 oocytes observed).

To test if BMK-1 was necessary for the formation of miniature anaphases, we utilized two *bmk-1* mutants: (1) a previously characterized allele, *bmk-1(ok391)*, that introduces a premature stop codon in the motor domain (*Bishop et al., 2005*), and (2) a new deletion of the entire *bmk-1* locus generated using CRISPR-Cas9 (*bmk-1(syb3914)*). To validate these deletions, we utilized IF imaging with an α-BMK-1 antibody and confirmed that BMK-1 was no longer present on the meiotic spindle in either mutant (no BMK-1 staining in 71/71 oocytes observed) (*Figure 7—figure supplement 1*). We then generated monopolar spindles, performed short-term Dynein AID, and performed IF in the *bmk-1(ok391)* background. Under these conditions, we were unable to observe any mini anaphase spindles (0/164 oocytes), even though monopole breakdown still occurred (*Figure 7B and C*, *Table 6*). This suggested that removal of BMK-1 function abolished microtubule reorganization and prevented chromosome segregation. To confirm this result, we generated a version of our Dynein AID live imaging strain (expressing GFP::tubulin and GFP::histone to visualize the spindle) containing the *bmk-1(syb3914)* deletion and performed acute dynein depletion coupled with ex utero imaging to determine if mini anaphases could still form. In *klp-18(RNAi)*, *bmk-1(syb3914)* worms in the absence of auxin, monopolar spindles formed and then chromosomes moved back towards the pole in a manner indistinguishable from normal monopolar anaphase (*Video 15*, n = 4). When auxin was added to deplete dynein, monopolar spindles broke down as expected and individual chromosomes remained associated with microtubule bundles. However, as time elapsed, there were no signs of segregation, microtubule density decreased, and chromosomes remained in the cytoplasm with no discernible anaphase spindles forming (*Figure 7D*, *Video 16*, n = 3). These data support the hypothesis that BMK-1 provides outward sorting force on microtubules during oocyte meiosis, redundant to the forces produced by KLP-18.

## Discussion

### Contributions of dynein to oocyte meiosis

Collectively, these data have contributed to a more complete model for how acentrosomal spindles are assembled and stabilized during oocyte meiosis (*Figure 8*). Dynein activity is required throughout the meiotic divisions to establish and maintain focused poles, and its removal via acute AID generated spindle defects within minutes. Dynein depletion increased the length of bipolar oocyte spindles and led to increased dispersion of microtubule minus ends across the spindle. When performing these depletions on a monopolar spindle, the monopole completely broke apart and microtubule bundles were ejected into the cytoplasm. Together, these findings demonstrate that dynein is required to stitch minus ends together into a stable pole structure.

Remarkably, we also found that after the monopole disassembled, microtubules were able to reorganize into a 'miniature' spindle; chromosomes then segregated on these spindles in the absence of dynein. This finding seemingly contradicts one of our previous studies, where we proposed that dynein could drive chromosomes towards poles along laterally associated microtubule bundles, representing a type of 'anaphase A' segregation mechanism (*Muscat et al., 2015*). Dynein was a good candidate for such a role since we found that it provided chromosomes with a minus-end-directed force prior to anaphase, and we

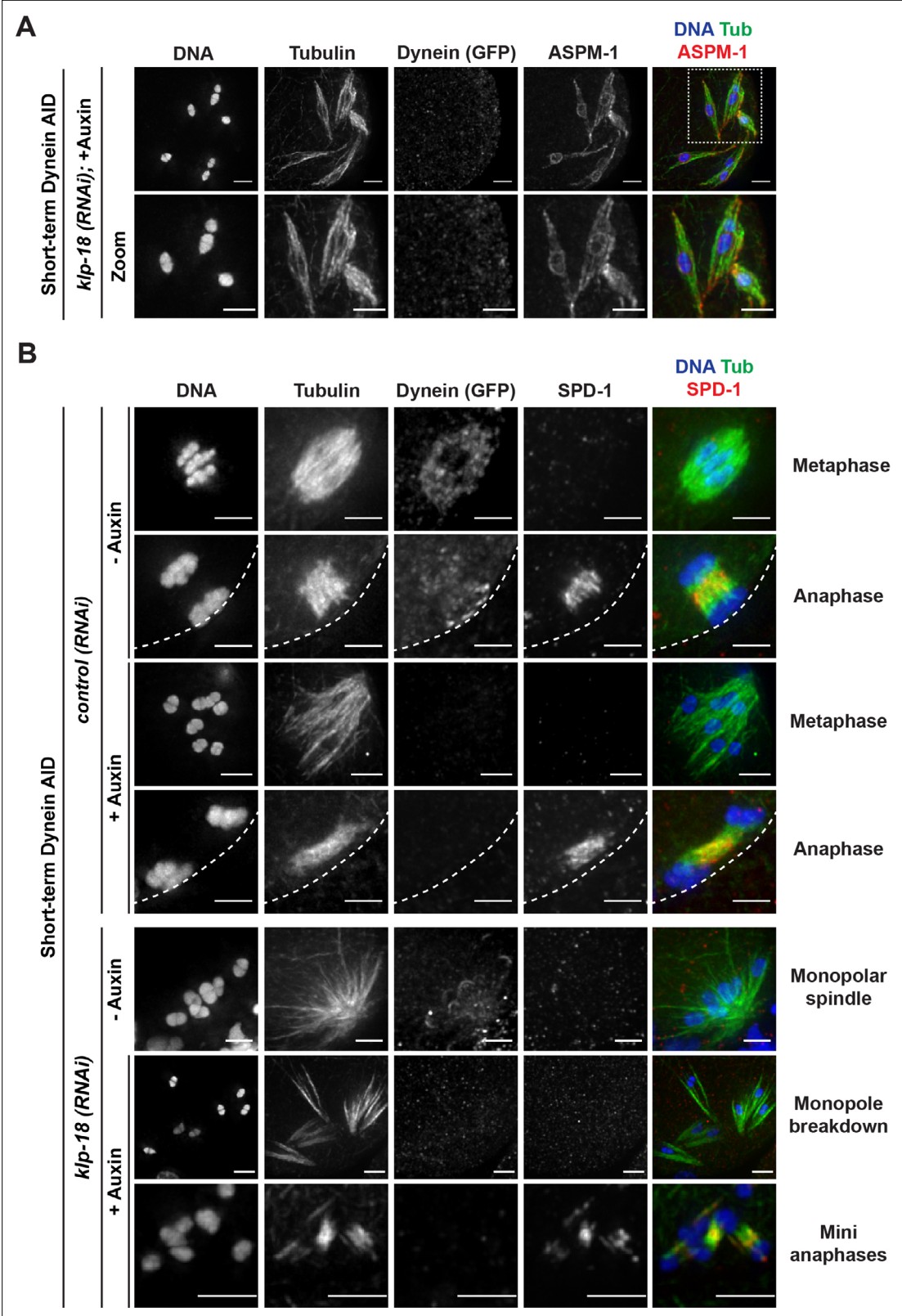

**Figure 6.** Microtubules reorganize into miniature bipolar spindles that can segregate chromosomes. (**A**) Immunofluorescence (IF) imaging of microtubules (green), DNA (blue), ASPM-1 (red), and dynein (not shown in merge) in monopolar spindle breakdown conditions (*klp-18(RNAi)* + short-term Dynein auxin-inducible degron [AID]). Microtubule bundles appear to reorganize around individual chromosomes, seen through ASPM-1 flanking either side of the chromosome; note that in these images ASPM-1 also appears to be on chromosomes, but that is background staining that sometimes

*Figure 6 continued on next page*

Figure 6 continued

occurs with this antibody (*Wignall and Villeneuve, 2009*) and is not real signal. (**B**) IF imaging of SPD-1 localization in the Dynein AID strain in control RNAi conditions (rows 1–4) or following *klp-18(RNAi)* (rows 5–7); shown are tubulin (green), DNA (blue), SPD-1 (red), and dynein (not shown in merge). SPD-1 does not localize to spindles in metaphase (rows 1, 3; 18/18 metaphases), but localizes to overlapping microtubules in anaphase spindles in the presence or absence of dynein (rows 2, 4; 17/17 anaphases). SPD-1 is not localized to monopolar spindles either before or after monopole breakdown (rows 5, 6; 60/60 monopoles and breakdowns), but can clearly be seen localized to miniature anaphases (row 7; 27/27 mini anaphases). Cortex is represented by the dashed line. All scale bars = 2.5 μm.

observed lagging chromosomes in anaphase following dynein inhibition (*Muscat et al., 2015*); lagging chromosomes upon dynein depletion have also been reported by others (*McNally et al., 2016*) and were observed in the current study (*Figure 2—figure supplement 3*). However, since chromosome movement was not completely blocked following dynein inhibition, when we proposed our original model we speculated that other mechanisms, such as spindle elongation, might also contribute to segregation (*Muscat et al., 2015*). Indeed, in subsequent years it has become clear that this anaphase-B-like mechanism is the major driver of chromosome segregation in *C. elegans* oocytes, and that anaphase A chromosome-to-pole movement provides only a minor contribution (*McNally et al., 2016*; *Laband et al., 2017*). Given these follow-up studies, it is now clear that if dynein plays a role in chromosome segregation, its contribution is redundant with other, more dominant, mechanisms; this would explain why most chromosomes were able to move poleward following dynein inhibition in our original study and in subsequent studies by other labs (*Muscat et al., 2015*; *McNally et al., 2016*; *Laband et al., 2017*; *Danlasky et al., 2020*). This updated view is also consistent with the results of our current study; in our Dynein AID depletions, unarrested bipolar spindles progress through anaphase and mini spindles formed following *klp-18(RNAi)* also undergo anaphase-like segregation, reaching comparable segregation distances to that of normal anaphase spindles. Interestingly, our studies also revealed that anaphase-B-like spindle elongation can occur in the absence of KLP-18. Although it is possible that KLP-18 may contribute to outward sliding of microtubules during wild-type anaphase, our work shows that this motor is not absolutely required for anaphase spindle elongation and demonstrates that there must be other factors that can perform this function.

## Multiple motors cooperate in *C. elegans* oocyte meiosis to effectively form and stabilize an acentrosomal bipolar spindle

Our work also accentuates the importance of balanced forces within a bipolar meiotic spindle; we found that loss of dynein activity had strong phenotypes that manifested rapidly, within 2 min of auxin treatment. Notably, acute dynein depletion caused spindle lengthening, suggesting that dynein may normally provide an inward force on the spindle, as has been demonstrated in previous studies of mitosis (*Ferenz et al., 2009*; *van Heesbeen et al., 2014*). Recently, our lab also demonstrated that inactivation of KLP-18 in stable, bipolar spindles caused spindle shortening followed by collapse of microtubule minus ends into a monopolar spindle (*Wolff et al., 2021*). This collapse is due to a loss of the outward force provided by KLP-18, theoretically enabling the inward force provided by dynein to dominate and highlighting how quickly an imbalance of these motor forces can lead to gross defects in spindle organization. In the current study, removing dynein from oocytes that already lacked KLP-18 resulted in a catastrophic breakdown of the monopolar spindle, releasing individual chromosomes into the cytoplasm. Going forward, it would be interesting to acutely remove these proteins at the same time to see whether oocyte spindles are able to maintain bipolarity and directly test whether these motors antagonize each other in *C. elegans*.

While KLP-18/kinesin-12 provides the major outward force in *C. elegans* oocyte meiosis (*Wolff et al., 2016*), this critical function is performed by kinesin-5 in mouse oocytes (*Schuh and Ellenberg, 2007*) and in mitosis in many organisms (reviewed in *Mann and Wadsworth, 2019*). Previous studies have shown that dynein and kinesin-5 antagonize each other in these organisms; inhibition of both motors simultaneously enables bipolar spindle formation (*Mitchison et al., 2005*; *Tanenbaum et al., 2008*; *Ferenz et al., 2009*; *van Heesbeen et al., 2014*). Interestingly, it has been shown that Kif15/kinesin-12 is capable of providing a supplemental outward force that can support bipolarity when kinesin-5 is inhibited (*Tanenbaum et al., 2009*; *Vanneste et al., 2009*; *Raaijmakers et al., 2012*; *Sturgill and Ohi, 2013*; *Sturgill et al., 2016*). Here, we have provided evidence that these roles have been reversed in *C. elegans* oocyte meiosis; BMK-1/kinesin-5 appears to be providing a redundant

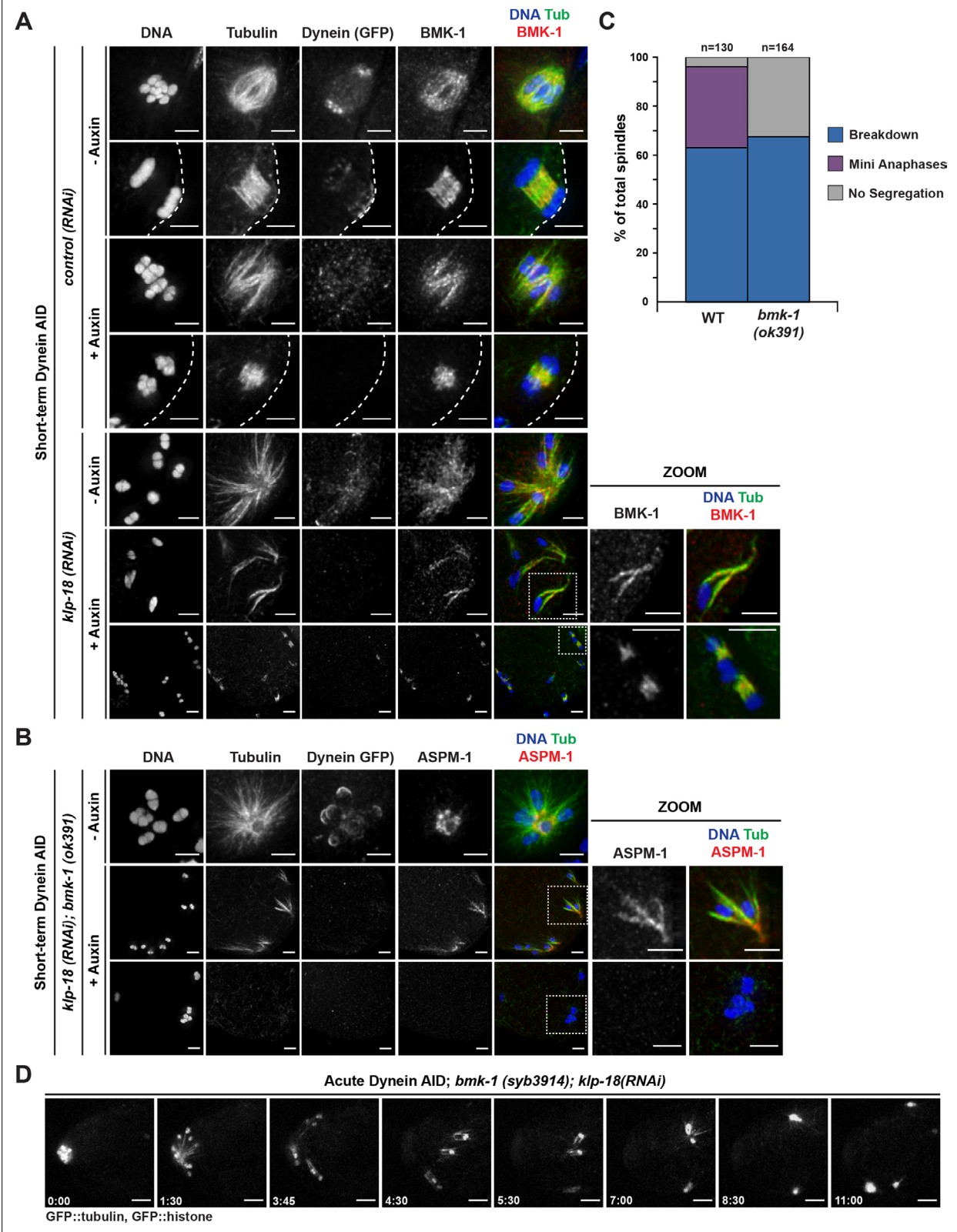

**Figure 7.** BMK-1 localizes to the meiotic spindle and is required for microtubule reorganization and the formation of miniature anaphases. (**A**) Immunofluorescence (IF) imaging of oocytes in either control or *klp-18(RNAi)* conditions in the Dynein auxin-inducible degron (AID) strain in the presence and absence of short-term auxin treatment; shown are tubulin (green), DNA (blue), BMK-1 (red), and dynein (not shown in merge). BMK-1 is localized to spindle microtubules in all conditions (52/52 metaphases, 43/43 anaphases), including following monopolar spindle breakdown (40/40

*Figure 7 continued on next page*

*Figure 7 continued*

monopoles and breakdowns) and in mini anaphases (15/15 mini anaphases). Cortex is represented by the dashed line. Scale bars = 2.5 µm. (**B**) IF imaging of embryos following *klp-18(RNAi)* in the Dynein AID strain lacking functional BMK-1 (*bmk-1(ok391)*); shown are tubulin (green), DNA (blue), ASPM-1 (red), and dynein (not shown in merge). Following monopolar spindle breakdown in the presence of auxin (rows 2, 3), embryos do not contain miniature anaphases and lack chromosome segregation. Scale bars = 2.5 µm. (**C**) Quantifications of images shown in (**B**) compared to wild-type (WT) embryos; monopolar spindles still break down, but no miniature anaphases are observed in embryos lacking BMK-1 function. (**D**) Ex utero live imaging of GFP::tubulin and GFP::histone following acute auxin treatment to remove dynein in *klp-18(RNAi); bmk-1(syb3914)* conditions; miniature anaphases do not form in the absence of BMK-1. Time elapsed shown in min:s. Scale bars = 5 µm. Validation of *bmk-1* mutants via IF imaging can be seen in *Figure 7—figure supplement 1*.

The online version of this article includes the following figure supplement(s) for figure 7:

**Figure supplement 1.** Immunofluorescence (IF) imaging validation of BMK-1 deletion in worm strains.

**Table 6.** Exact numbers and percentages from quantification of IF images, to determine if the removal of functional BMK-1 prevents the formation of miniature anaphases in KLP-18/dynein-depleted oocytes (corresponds to *Figure 7C*).

| Condition | Treatment | Breakdown | Mini anaphases | No segregation |
|---|---|---|---|---|
| *klp-18 (RNAi)* | *bmk-1 (WT)* | 82/130 (63.1%) | 43/130 (33.1%) | 5/130 (3.8%) |
| | *bmk-1 (ok391)* | 111/164 (67.7%) | 0/164 (0.0%) | 53/164 (32.3%) |

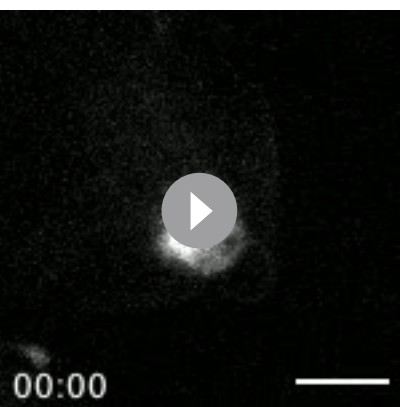

**Video 15.** Monopolar spindle dynamics are normal in the absence of BMK-1.
Shows a Dynein AID; *bmk-1(syb3914)* oocyte expressing GFP::tubulin and GFP::histone following *klp-18(RNAi)* to generate monopolar spindles, dissected into control Meiosis Medium solution. Without functional BMK-1 or KLP-18, monopolar spindles can form and monopolar anaphase still occurs, with chromosomes moving outwards in metaphase and back towards the monopole in anaphase. Phenotypes are consistent across all oocytes filmed (n = 4). Time elapsed shown in min:s. Scale bar = 5 µm.
https://elifesciences.org/articles/72872/figures#video15

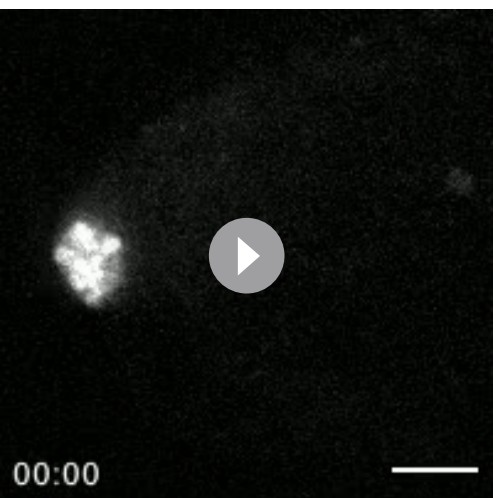

**Video 16.** Loss of BMK-1 function prevents the formation of miniature anaphases.
Shows a Dynein AID; *bmk-1(syb3914)* oocyte expressing GFP::tubulin and GFP::histone following *klp-18(RNAi)* dissected into Meiosis Medium containing 100 µM auxin to deplete dynein. Without functional BMK-1, formation of miniature bipolar anaphases does not occur after the breakdown of the monopolar spindle. Corresponds to Figure 6D. Phenotypes are consistent across all oocytes filmed (n = 3). Time elapsed shown in min:s. Scale bar = 5 µm.
https://elifesciences.org/articles/72872/figures#video16

outward sorting force that was only detectable once we had removed KLP-18 and dynein from the meiotic spindle. Why the roles of kinesin-5 and kinesin-12 have been seemingly switched between different organisms is a fascinating open question. In the future, it would be interesting to perform KLP-18/DHC-1 double depletions in a strain lacking BMK-1 function to further probe the relationship between these three motors during acentrosomal spindle formation in *C. elegans*.

While this work has expanded our understanding of the role BMK-1 plays in *C. elegans* oocyte meiosis, further experimentation will be valuable for understanding the exact mechanism of BMK-1 function in the context of a normal, bipolar meiotic spindle. Biophysical assays to determine motor walking speed and force generation would help frame how much BMK-1 actually contributes in comparison to other meiotic motors such as KLP-18. Also, since kinesin-5 activity is known to be regulated in other systems by kinases and protein-protein interactions (reviewed in *Mann and Wadsworth, 2019*), it would be beneficial to determine if BMK-1 has any interacting partners that provide some regulation of function. Aurora B kinase (AIR-2) has been shown to be required for BMK-1 spindle localization and AIR-2 can phosphorylate BMK-1 in vitro (*Bishop et al., 2005*) implicating AIR-2 in BMK-1 regulation. Inhibition of AIR-2's kinase activity also leads to collapsed oocyte spindles (*Divekar et al.,*

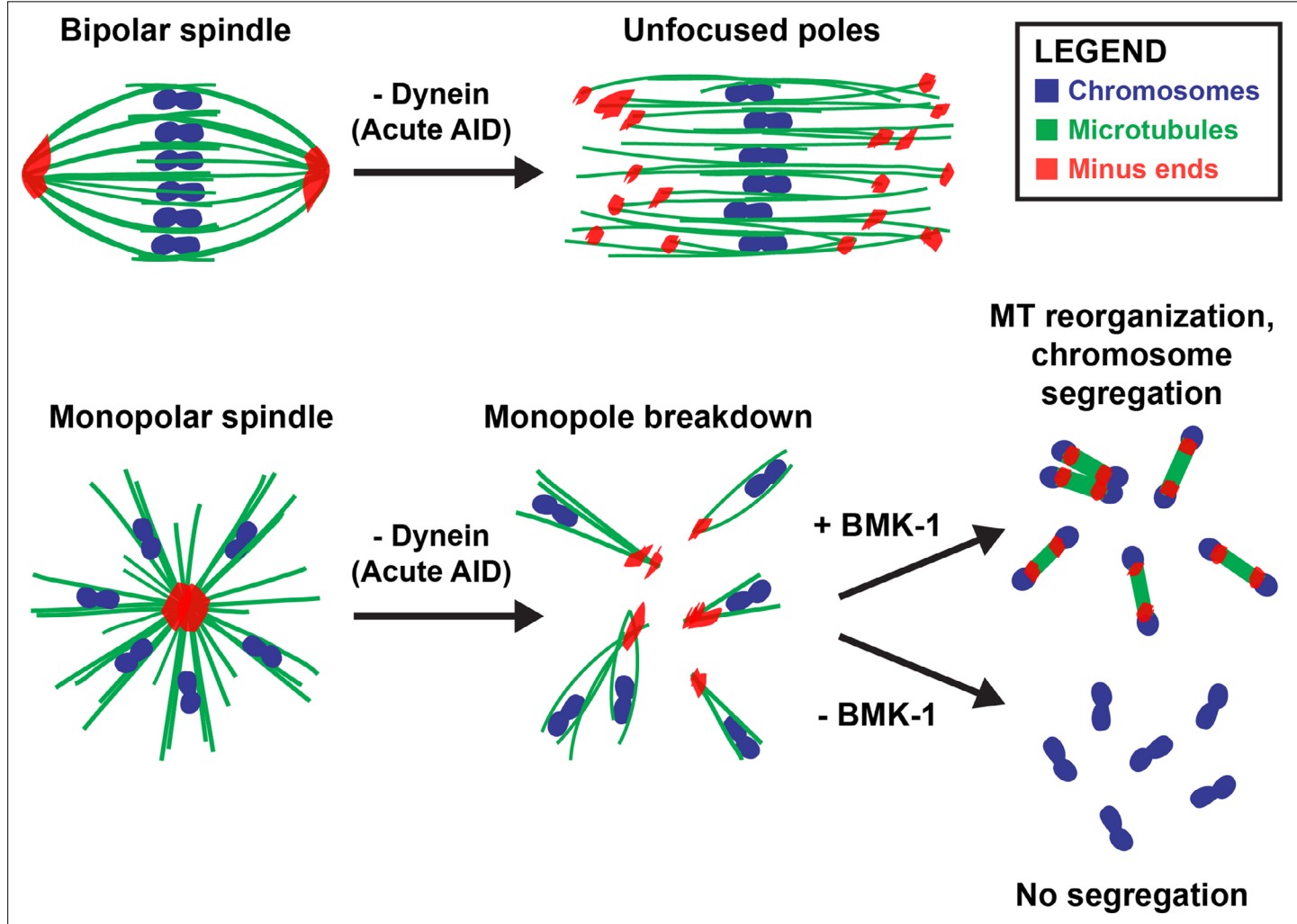

**Figure 8.** Dynein, KLP-18, and BMK-1 work in concert to establish and maintain spindle bipolarity in *C. elegans* oocyte meiosis. Chromosomes (blue), microtubules (green), and microtubule minus ends (red). Dynein is required throughout the meiotic divisions to maintain focused acentrosomal poles. If removed from stable bipolar spindles (top) using short-term depletions, poles rapidly unfocus and splay, leading to the same phenotype as long-term depletions. Dynein depletion from monopolar spindles (bottom) ejects individual chromosomes and associated microtubule bundles into the cytoplasm. BMK-1 is able to provide an outward sorting force in the absence of KLP-18 and dynein, enabling reorganization of microtubules into a miniature anaphase spindle, promoting chromosome segregation. In the absence of BMK-1, these miniature anaphases cannot form and anaphase-like segregations no longer occur.

*2021a*), consistent with a role for AIR-2 in regulating spindle force generation. However, it is likely that other factors also regulate BMK-1. Finally, another interesting question is whether BMK-1 acts to reorganize the existing microtubule bundles that remain chromosome-associated following mono-pole breakdown or whether chromatin-mediated microtubule nucleation also plays a role in miniature spindle formation, as it does in many systems during acentrosomal spindle assembly (*Meunier and Vernos, 2016*; *Radford et al., 2017*).

## Different acentrosomal pole proteins have distinct roles in pole coalescence and stability

Our analysis of ASPM-1 and LIN-5 provides further evidence that these proteins are directly influencing acentrosomal pole organization and stability by directing cytoplasmic dynein localization. In either *aspm-1* or *lin-5* RNAi, spindle phenotypes were nearly identical to those seen in Dynein AID experimentation, and co-depletion of dynein did not worsen those phenotypes. The shared phenotype across depletions of these proteins suggests that these pole proteins function in a single pathway; this provides some credence to the concept that acentrosomal poles could contain a cross-linked network of interacting pole proteins that provide cohesion and stability. However, recent work from our lab has made it clear that not all pole protein depletions yield the same defects in spindle bipolarity. Acute depletion of ZYG-9, a homolog of the microtubule polymerase XMAP215, disrupts spindle integrity in a manner that does not resemble Dynein AID depletions. While dynein depletion splays spindle poles, microtubule bundles appear to remain stable and unperturbed, generating a rectangular spindle that can still progress through meiosis (albeit with spindle rotation defects). However, acute ZYG-9 depletion leads to a unstable multi-polar spindle with defects in the middle of the spindle as well as at the poles, suggesting a broader role of ZYG-9 (*Mullen et al., 2022*).

From these recent studies, it is clear that multiple groups of proteins are required to establish and maintain acentrosomal poles. However, despite having similar localizations in the meiotic spindle, pole proteins have separable functions that impact pole stability in unique ways. Rather than all pole proteins interacting as a cohesive, cross-linked network around microtubule minus ends, it seems more plausible that different groups of proteins work in parallel to form and stabilize acentrosomal poles. These degron-based depletions with essential pole proteins have allowed for further mechanistic understanding of how each pole protein is contributing to spindle stability. There remain numerous pole proteins that could be subjected to rapid depletion via the AID system; the ability to probe the roles of essential proteins in both spindle assembly and maintenance is vital to building a more robust model of how meiotic spindles are able to achieve bipolarity in the absence of centrosomes to faithfully congress and segregate chromosomes.

## Materials and methods
### *C. elegans* strains used

| Name | Description | Genotype |
|------|-------------|----------|
| CA1215 | *dhc-1::degron::GFP; P$_{sun-1}$::TIR1::mRuby* | *dhc-1(ie28[dhc-1::degron::GFP]) I; ieSi38 IV* |
| PHX3914 | *dhc-1::degron::GFP; P$_{sun-1}$::TIR1::mRuby; GFP::H2B; GFP::tubulin; Δbmk-1* | *dhc-1(ie28[dhc-1::degron::GFP]) I; unc-119(ed3) III; ruIs32[unc-119(+) pie-1::gfp::h2b] III; ruIs57[unc-119(+) pie-1::GFP::tubulin]; ieSi38 IV; bmk-1(syb3914) V* |
| SMW22 | *P$_{sun-1}$::TIR1::mRuby; mCherry::tubulin* | *unc-119(ed3) III; weIs21 [pJA138 (pie-1::mCherry::tubulin::pie-1)]; ieSi38 IV* |
| SMW31 | *P$_{sun-1}$::TIR1::mRuby; GFP::H2B; GFP::tubulin* | *unc-119(ed3) III; ruIs32[unc-119(+) pie-1::gfp::h2b] III; ruIs57[unc-119(+) pie-1::GFP::tubulin]; ieSi38 IV* |
| SMW46 | *dhc-1::degron::GFP; P$_{sun-1}$::TIR1::mRuby; GFP::H2B; GFP::tubulin* | *dhc-1(ie28[dhc-1::degron::GFP]) I; unc-119(ed3) III; ruIs32[unc-119(+) pie-1::gfp::h2b] III; ruIs57[unc-119(+) pie-1::GFP::tubulin]; ieSi38 IV* |
| SMW47 | *dhc-1::degron::GFP; P$_{sun-1}$::TIR1::mRuby; bmk-1(ok391)* | *dhc-1(ie28[dhc-1::degron::GFP]) I; ieSi38 IV; bmk-1(ok391) V* |

*Continued on next page*

*Continued*

| Name | Description | Genotype |
|------|-------------|----------|
| SMW48 | *dhc-1::degron::GFP; P*<sub>*sun-1*</sub>*::TIR1::mRuby mCherry::tubulin* | *dhc-1(ie28[dhc-1::degron::GFP]) I; weIs21 [pJA138 (pie-1::mCherry::tub::pie-1)]; ieSi38 IV* |
| SV1005 | *bmk-1(ok391)* | *bmk-1(ok391) V* |

## Generation of *C. elegans* strains

CA1215 was generated via CRISPR/Cas9 editing of the endogenous *dhc-1* locus (*Zhang et al., 2015*). PHX3914 was generated via CRISPR/Cas9 editing of the endogenous *bmk-1* locus by SunyBiotech. SV1005 was a knockout strain generated by the *C. elegans* Deletion Mutant Consortium. SMW46-SMW48 were generated by crossing two strains and screening multiple generations of progeny to ensure homozygosity of all desired traits.
SMW46: Crossed males of SMW31 with CA1215 hermaphrodites
SMW47: Crossed males of CA1215 with SV1005 hermaphrodites
SMW48: Crossed males of CA1215 with SMW22 hermaphrodites

## RNAi feeding

Our RNAi protocol is described in detail in *Wolff et al., 2022*. Briefly, from an RNAi library (*Kamath et al., 2003*), individual RNAi clones were picked and grown overnight at 37°C in LB with 100 µg/mL ampicillin. Overnight cultures were spun down, resuspended, and plated on nematode growth medium (NGM) plates containing 100 µg/mL ampicillin and 1 mM IPTG. Plates were dried in a dark space overnight at 22°C; at the same time, worms were synchronized for experimentation via bleaching of gravid adults, collecting remaining embryos, and plating on foodless plates to hatch overnight. The next day, newly hatched L1 worms were transferred to appropriate RNAi plates and grown to adulthood at 15°C for 5–6 days. In specific experiments utilizing *lin-5(RNAi)* or *aspm-1(RNAi)*, L1 worms were plated on empty vector (EV) L440 control RNAi plates for 2–3 days, then transferred to *lin-5(RNAi)* or *aspm-1(RNAi)* plates 72 hr prior to fixation.

## Immunofluorescence and antibodies

Our IF protocol is described in detail in *Wolff et al., 2022*. Briefly, adult worms, grown on either EV control RNAi or experimental RNAi, were picked into a 10 µL drop of Meiosis Medium (0.5 mg/mL Inulin, 25 mM HEPES, and 20% FBS in Leibovitz's L-15 Media) (*Laband et al., 2018*) in the center of a poly-L-lysine-coated glass slide. Worms were dissected, then fixed via freeze cracking and plunging into –20°C MeOH as described in *Oegema et al., 2001*. Oocytes and embryos were fixed for 40–45 min, rehydrated in PBS, and blocked in AbDil (PBS with 4% BSA, 0.1% Triton-X-100, 0.02% NaN₃) overnight at 4°C. Primary antibodies were diluted in AbDil and incubated with sample overnight at 4°C. The following day, the samples were moved to room temperature and rinsed three times in PBST (PBS with 0.1% Triton-X-100), then incubated with secondary antibodies (diluted in PBST) for 2 hr. Next, the samples were washed three times in PBST again and incubated with mouse anti-α-Tubulin-FITC (diluted in PBST) for 2 hr. Again, the samples were washed three times in PBST, then incubated with Hoescht (1:1000 in PBST) for 15 min. Finally, the samples were washed two times in PBST, mounted in 0.5% *p*-phenylenediamine, 20 mM Tris-Cl, pH 8.8, 90% glycerol, then sealed with nail polish and stored at 4°C.

The primary antibodies used in this study were rabbit-α-ASPM-1 (1:5000, gift from Arshad Desai), rabbit-α-BMK-1 (1:250, gift from Jill Schumacher), mouse-α-Tubulin-FITC (1:500, DM1α, Sigma), mouse-α-GFP (1:250, 3E6, Invitrogen), mouse-α-SUMO (1:500, gift from Federico Pelisch), rabbit anti-SEP-1 (1:250, gift of Andy Golden), and rabbit-α-SPD-1 (*Mullen et al., 2017*). All rabbit and mouse Alexa Fluor secondary antibodies (Invitrogen) were used at 1:500.

## Ex utero live imaging

Fifteen adult worms, grown on either control RNAi or experimental RNAi, were picked into a 10 µL drop of Meiosis Medium (described above) in the center of a custom-made apparatus for live imaging (*Laband et al., 2018*; *Divekar et al., 2021b*). All worms were quickly dissected, and an eyelash pick was used to push remaining worm bodies to the outside of the drop, leaving only the oocytes and

embryos in the center to avoid disruption from worm movement during the imaging process. Vaseline was laid in a ring around the drop through a syringe, and a 18 × 18 mm #1 coverslip was laid on top of the Vaseline ring, sealing the drop. This sealed slide was moved immediately to the spinning disk stage and inverted, allowing oocytes and embryos to float down to the surface of the coverslip and be subsequently imaged. For movies of unarrested spindles, we looked for embryos that had extruded the first polar body but where the meiosis II spindle had not yet formed; thus t = 0 was synchronized to the onset of meiosis II. For experiments involving *emb-30(RNAi)*, t = 0 was defined as the first frame that we were able to capture after finding a bipolar spindle to image. For these experiments, we did not image spindles that exhibited hallmarks of prolonged metaphase arrest when they were dissected (e.g., stretching of chromosomes and widening of the spindle midzone) to avoid artifacts arising from the metaphase-arrest condition.

## Microscopy

All fixed imaging was performed on a DeltaVision Core deconvolution microscope with a 100x objective (NA = 1.4) (Applied Precision). This microscope is housed in the Northwestern University Biological Imaging Facility supported by the NU Office for Research. Image stacks were obtained at 0.2 μm z-steps and deconvolved using SoftWoRx (Applied Precision). All IF images in this study were deconvolved and displayed as full maximum intensity projections of data stacks encompassing the entire spindle structure (typically ~4–6 μm).

All live imaging was performed using a spinning disk confocal microscope with a 63x HC PL APO 1.40 NA objective lens. A spinning disk confocal unit (CSU-X1; Yokogawa Electric Corporation) attached to an inverted microscope (Leica DMI6000 SD) and a Spectral Applied Imaging laser merge ILE3030 and a back-thinned electron-multiplying charge-coupled device (EMCCD) camera (Photometrics Evolve 521 Delta) were used for image acquisition. The microscope and attached devices were controlled using Metamorph Image Series Environment software (Molecular Devices). Typically, 10–15 z-stacks at 1 μm increments were taken every 15–30 s at room temperature. Images were processed using ImageJ; images are shown as maximum intensity projections of the entire spindle structure. The spinning disk microscope is housed in the Northwestern University Biological Imaging Facility supported by the NU Office for Research.

## Auxin treatment

Auxin treatments were performed using three different methods (long term, short term, and acute); further details and diagrams of these methods can be found in *Divekar et al., 2021b*.

### Long-term AID

Standard NGM RNAi plates were prepared, but 500 mM auxin (dissolved in 100% EtOH) was added to a final concentration of 1 mM prior to pouring 6 cm plates. These plates were kept in a dark room to solidify and dry. For long-term auxin experiments, auxin plates were seeded with the desired RNAi culture, incubated overnight to induce dsRNA production, and put at 4°C until needed. In order to achieve a 4 hr auxin treatment, adult worms were grown on separate RNAi plates that did not contain auxin (poured and seeded in parallel with the auxin plates), then transferred to auxin-containing plates 4 hr prior to EtOH fixation or dissection and IF.

### Short-term AID

A solution of Meiosis Medium containing 1 mM auxin was prepared from a 200 mM auxin stock (dissolved in 100% EtOH) and kept on ice. Adult worms were picked into 10 μL drops of Medium + auxin, and then the slides were placed inside a custom-made humidity chamber to avoid evaporation of the drop. Following incubation for the desired time (30–45 min), worms were dissected and subjected to the standard IF protocol described above.

### Acute AID

For acute auxin treatments coupled with ex utero live imaging, a solution of Meiosis Medium containing 100 μM auxin was made from a 200 mM auxin stock (dissolved in 100% EtOH). Adult worms were picked into 10 μL drops of Medium + auxin and immediately dissected on a custom-made live imaging

slide apparatus (as described in the ex utero imaging section). Slides were moved quickly to the spinning disk microscope, and acquisition was started as soon as an oocyte could be found.

## Ethanol fixation for germline counting

Ethanol fixation and germline counting methods are described in more detail in *Wolff et al., 2022*. Briefly, SMW46 worms were subjected to EV control RNAi (or *klp-18(RNAi)*), utilizing methods described above, and were grown to adulthood over 5 days. 4 hr prior to fixation, worms were either transferred to EV control RNAi (with 1 mM auxin) or left on their original RNAi plates. For each biological replicate, ~40 adults were picked off their respective plates into a 10 µL drop of M9 on a standard glass slide, and a small piece of Whatman paper was used to absorb excess M9, with care being used to avoid pulling worms onto filter paper. Once worms had formed a tight cluster with little residual M9, a 10 µL drop of 100% EtOH was quickly pipetted onto the worms. Within seconds, worms became rigid and straight; once the first EtOH drop had dried, another 10 µL drop was applied and allowed to completely dry. After the third drop was added and dried, a 10 µL drop of 50% Vectashield Mounting Media (Vector Laboratories H-1000) and 50% M9 was placed onto the worms, and a 18 × 18 mm coverslip was gently placed on top. Excess media was aspirated away, slides were sealed with nail polish, and stored at 4°C (typically imaged within a week of fixation).

## Data analysis

*Figure 1D*: Quantifications were made by viewing whole Dynein AID worms expressing GFP::tubulin and GFP::histone and observing three positions in each worm (–1 position, Spermatheca, + 1 position) in both gonad arms. When spindle morphology was clear enough to confidently categorize, spindles were then counted into one of four categories (Microtubule Cage, Unfocused, Focused, Anaphase). The number of spindles counted across all slides (n) is placed above each respective location and condition. Each bar comprises counts from at least four biological replicates. Exact numbers used to generate the graph are given in *Table 1*.

*Figure 1E and F*: Categorization of acentrosomal poles into either focused or unfocused/splayed was done by eye, looking at both microtubule and ASPM-1 channels. In any case where microtubule bundles were clearly separated from one another at an acentrosomal pole, with multiple ASPM-1 foci, that spindle was considered unfocused/splayed. Quantifications of acentrosomal pole splaying were made by categorizing multiple fixed images; spindles counted across all slides (n) are placed above each respective condition. Each bar comprises spindles imaged across three biological replicates. Exact numbers used to generate the graph in *Figure 1F* are given in *Table 2*.

*Figure 1—figure supplement 1*: Quantifications were made by viewing whole worms expressing GFP::tubulin and GFP::histone and TIR1, but lacking a degron-tagged version of DHC-1 (labeled as 'TIR1 control'), and observing three positions in each worm (–1 position, Spermatheca, and +1 position) in both gonad arms. When spindle morphology was clear enough to confidently categorize, spindles were then counted into one of four categories (Microtubule Cage, Unfocused, Focused, Anaphase). The number of spindles counted across all slides (n) is placed above each respective location and condition. Each bar comprises counts from at least three biological replicates. Exact numbers used to generate the graph are given in *Table 3*.

*Figure 1—figure supplement 2A*: Quantifications were made by viewing whole Dynein AID worms expressing GFP::tubulin and GFP::histone and observing two positions in each worm (Spermatheca and +1 position) in both gonad arms. When spindle morphology was clear enough to confidently categorize, spindles were then counted into one of three categories (Unfocused, Focused, Anaphase). The number of spindles counted across all slides (n) is placed above each respective location and condition. Each bar comprises counts from at least three biological replicates. Exact numbers used to generate the graph are given in *Table 4*.

*Figure 2B*: To determine the degree of splaying at acentrosomal poles, we employed a quantification that measures the ratio of spindle widths between the equator of the spindle (as defined by the metaphase-aligned chromosomes) with the average width of both poles (guided by ASPM-1 staining). Using this pole splaying ratio (r), we displayed these data as boxplots using RStudio; boxplots represent first quartile, median, and third quartile. Statistical significance between the mean *r* value of control and auxin-treated spindles was determined via a two-tailed *t*-test for both unarrested and metaphase-arrested conditions.

*Figure 2—figure supplement 1A and B*: The same data set of control and auxin-treated spindles measured in *Figure 2B* were further analyzed here. Using Fiji, a rectangular ROI (12 µm or 14 µm wide × 4 µm tall) was used on all images of a single condition to measure the average pixel intensity of ASPM-1 across the whole width of the ROI via the Plot Profile Tool. These values were then averaged from all images in that condition to produce a single value for ASPM-1 intensity at each pixel across the standardized 12 µm/14 µm length. This was plotted as a solid line for each condition, with shaded areas above and below the line representing the SEM. All images were taken with the same exposures for all channels. Spindles were sum projected over 20 slices (0.2 µm step size) for each image prior to measurement. Each condition was averaged using images from at least three biological replicates. Spindle length measurements were done in Imaris. Poles were projected into 3D volumes using the Surfaces Tool (based on fluorescence intensity), and the center of each volume was determined. The exact micron distance between the designated center of the two volumes was measured, and this distance was averaged across images of the same condition. Quantifications were arranged as boxplots, and statistical significance was determined via a two-tailed *t*-test. Each condition was averaged using images from at least three biological replicates.

*Figure 2—figure supplement 2B and C*: Timelapses of the TIR1 control strain were opened in Imaris to measure both spindle length and the ratio of pole splaying (r), as described for *Figure 2B*. Quantifications of spindle length and pole splaying (r) were performed as described above for *Figure 2B* and *Figure 2—figure supplement 1*. For display of this data, we employed two methodologies: to avoid averaging out individual variations, we first plotted all 10 timelapses as single traces (green hues for control, violet hues for auxin-treated). However, to get a better visualization of the trends over the course of the timelapse, we also generated averages of the change in length/pole ratio (r) over time (when compared to baseline). For average change plots, shaded area represents the standard deviation amongst all five timelapses in that condition.

*Figure 3A*: Quantifications were made by viewing whole Dynein AID worms expressing GFP::tubulin and GFP::histone and observing all +1 spindles. When spindle morphology was clear enough to confidently categorize, spindles were then counted as either monopolar (a single structure with chromosomes fanned away from the center) or undergoing breakdown (noticeable separation of microtubules and chromosomes). The number of structures counted across all slides (n) is placed above each respective location and condition. Each bar comprises counts from three biological replicates. Exact numbers used to generate the graph are given in *Table 5*.

*Figure 4C*: Timepoints from live imaging videos were analyzed utilizing Imaris. Chromosomes were rendered into 3D surfaces using the 'Surfaces' tool, and the center of each volume was determined. For each chromosome that was within the z-stack throughout the entirety of the timelapse, measurements of the distance between the center of segregating chromosomes were taken. These distances were averaged at each 15 s timepoint, and each timelapse was standardized to the same starting point (onset of anaphase) to allow for further averaging between separate videos. The average segregation distance across each condition was plotted (number of videos in each condition represented by n), and shaded area represents SEM of each average.

*Figure 7B and C*: Classification of monopolar spindles into one of three categories (Breakdown, Mini Anaphases, No Segregation) was done by eye, looking at DNA, microtubule, and ASPM-1 channels. In any case where some bivalents were clearly separated from the monopolar spindle, yet a distinct ASPM-1 monopole remained with some number of attached bivalents, was classified as 'Breakdown.' Whenever all six bivalents could be seen dispersed in the cytoplasm with no remaining ASPM-1 monopole, and appeared to have some segregation between chromosomes, that was considered 'Mini Anaphase.' Any oocytes containing dispersed chromosomes without any indication of segregation or an anaphase spindle were classified as 'No Segregation.' The number of structures counted across all slides (n) is placed above each respective condition. Each bar comprises counts from five biological replicates. Exact numbers used to generate the graph are given in *Table 6*.

## Acknowledgements

We thank all members of the Wignall lab and the WiLa ICB for support and discussion, and Emily Czajkowski, Nikita Divekar, Hannah Horton, Juhi Narula, and Ian Wolff for critical reading of the manuscript. Additionally, we thank Karlin Compton for their help in piloting certain experimental conditions. We thank SunyBiotech for their service in generating PHX3914 for live imaging experiments.

Finally, we thank Liangyu Zhang and the Dernburg Lab for providing CA1215 (the original *dhc-1::degron::GFP* worm strain) and for corresponding with us to provide advice, and Arshad Desai, Andy Golden, Federico Pelisch, and Jill Schumacher for antibodies. Some strains were provided by the Caenorhabditis Genetics Center (CGC), which is funded by NIH Office of Research Infrastructure Programs (P40 OD010440). This work was funded by NIH R01GM124354 (to SMW) and by NIH/NCI training grant T32 CA009560 (to GCM).

## Additional information

### Funding

| Funder | Grant reference number | Author |
| --- | --- | --- |
| National Institute of General Medical Sciences | R01GM124354 | Sarah M Wignall |
| National Cancer Institute | T32CA009560 | Gabriel Cavin-Meza |

The funders had no role in study design, data collection and interpretation, or the decision to submit the work for publication.

### Author contributions

Gabriel Cavin-Meza, Conceptualization, Data curation, Formal analysis, Investigation, Methodology, Validation, Visualization, Writing - original draft, Writing – review and editing; Michelle M Kwan, Investigation; Sarah M Wignall, Conceptualization, Data curation, Formal analysis, Funding acquisition, Methodology, Project administration, Supervision, Validation, Visualization, Writing – review and editing

### Author ORCIDs

Gabriel Cavin-Meza (ID) http://orcid.org/0000-0001-5820-4071
Sarah M Wignall (ID) http://orcid.org/0000-0001-9828-9356

### Decision letter and Author response

Decision letter https://doi.org/10.7554/eLife.72872.sa1
Author response https://doi.org/10.7554/eLife.72872.sa2

## Additional files

### Supplementary files

• Transparent reporting form

### Data availability

All data generated or analyzed in this study are included in the manuscript and supporting files. Source data files have been provided for Figure 2B, Figure 2 - figure supplement 1, Figure 2 - figure supplement 2, and Figure 4C.

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
