## [Editor Report]

Oocytes in most species contain acentrosomal spindles and how these are formed and maintained is not entirely clear. Cavin-Meza et al. find that dynein is required for the integrity of bipolar spindle poles as well as spindle monopoles induced by depletion of the kinesin-12 motor KLP-18. They further show that in the absence of dynein motors, and functional spindle poles, microtubules are still able to organize into spindle-like structures via the action of chromosomes and the kinesin-5 motor BMK-1. They also found that chromosomes can still be segregated in an anaphase-like motion in these oocytes.

---

## [Decision Letter]

**Decision letter after peer review:**

Thank you for submitting your article "Multiple motors cooperate to establish and maintain acentrosomal spindle bipolarity in *C. elegans* oocyte meiosis" for consideration by *eLife*. Your article has been reviewed by 4 peer reviewers, including Federico Pelisch as Reviewing Editor and Reviewer #1, and the evaluation has been overseen by Jessica Tyler as the Senior Editor. The following individual involved in review of your submission has agreed to reveal their identity: Thomas Müller-Reichert (Reviewer #2).

Essential revisions:

While all reviewers found your work very interesting, there was agreement that a number of important issues need to be addressed in a revised version. The following is a summary of the main concerns that have to be addressed and you will find the detailed reviews below.

1) The phenotypes should be fully analysed and described quantitatively as much as possible. For each statement, the measured values, the number of measurements and the statistical data should be clearly provided also in the text.

2) The newly generated AID strain(s) need to be more thoroughly characterised:

a) Quantitative information on dynein depletion and its comparison with previously used methods (RNAi and ts alleles) is needed;

b) it would also be important to make sure none of the observed effects depend on the AID or AID::GFP tag which can frequently result in a partially defective protein.

c) control that auxin alone (without any AID) does not cause splayed poles in exposed isolated oocytes.

3) Assessment of protein depletions should be presented, more so in cases where a 'negative' result is shown (see the examples mentioned in the review reports regarding lin-5 and aspm-1).

4) Data in the present manuscript, as it stands, shows that the lab's previous suggested model of Dynein action on chromosome movement is compromised. Also, Dynein localisation itself is different in some cases when compared to the Muscat et al., (2015) paper. Without proper discussion of this issue including plausible explanations for this discrepancy, the reader is left wondering what the actual functions of Dynein are.

5) Results should be based on live imaging experiments whenever possible and the quality of the videos needs to be improved. For example, Figure 2 – supplement 2 is a key Figure and these videos should start with properly assembled Metaphase I spindles and from there, pole disassembly should be assessed. Also, as presented, the difference in intensity between – and + auxin makes this result not entirely convincing to me.

6) Unpublished data should be documented within this manuscript or not discussed at all.

*Reviewer #1:*

Oocytes in most species contain acentrosomal spindles and the understanding of how these spindles are assembled is limited. The manuscript by Cavin-Meza et al., therefore addresses an important question. The manuscript describes a set of different phenotypes after depletion of the motor proteins Dynein, KLP-18, and BMK-1. While the role of these proteins has been addressed before with variable depth, the current manuscript shows novel phenotypes revealing potential novel functions for these proteins on acentrosomal spindle assembly/stability.

The first part of the manuscript focuses on the role of dynein in female spindle stability. While the use of the AID::dynein allele is an elegant way to address spindle stability, there is insufficient evidence to fully back the authors' claims. A more detailed comparison between this AID allele and the methods used before to assess dynein depletion/loss of function would be critical. Particularly, a comparison with a ts allele, which can also be (and indeed was) used for fast dynein inactivation would increase the confidence in the observed results. This is linked to the apparent lack of reproducibility of some data included in the Muscat et al., (2015) paper from the same lab. It is not clear what the authors' model is for the two different roles (pole stability and chromosome segregation) they propose for Dynein and whether different dynein populations control these roles (spindle and kinetochore).

The authors take advantage of arresting oocytes at metaphase I to allow spindles to form before the acute depletion of Dynein. This is a nice system because it would allow a proper dynamic analysis of spindle pole disassembly in vivo. However, it appears that 'anecdotal' information is taken from the few live imaging experiments and most of the conclusions are taken from experiments involving fixed samples.

Overall, the manuscript includes a set of (very) interesting phenotypes, but it does not go a long way in their characterisation or mechanisms involved. Therefore, in order to thoroughly characterise the presented phenotypes and to put them in context with existing data, the manuscript would benefit from extensive revisions, including new experiments and quantification of all the phenotypes described.

Recommendations for the authors

– The phenotypes should be fully analysed and described quantitatively as much as possible: tubulin levels, spindle length, spindle width, etc.

– The need for live imaging as the core basis for the authors' claims is exemplified in this figure. While the authors do include data from live imaging experiments which seem promising, there are several issues with these experiments that require attention.

1) The authors have an extremely useful system to analyse disassembly but need to make full use of it. They need make sure they image the newly fertilised oocyte (first one after the spermatheca), therefore avoiding side effects due to prolonged arrest.

2) It is not clear how the authors define t = 0. This is not a trivial issue, and it is vital to know whether the two oocytes being analysed have the same 'history' (i.e. been arrested for ~ same time). For example, if one oocyte has just undergone fertilisation whereas the other has been arrested for longer, susceptibility to Dynein depletion could be drastically altered.

3) The lack of consistency in the signal intensity is concerning (see Figure 2D, GFP::histone signal differs massively between '- Auxin' and '+ Auxin').

4) In Figure 2 – supplement 2, the issue highlighted in 3) is even more concerning, but additionally, and equally or more important, the authors should be able to start the imaging experiment from a 'normal' metaphase in both conditions. Without this careful comparison (in addition to imaging GFP::ASPM-1 as described below) the claims about the pole stability defects are not entirely clear.

5) Why is there no Dynein signal on kinetochores as reported previously? Is this related to fixation? In fact, In Figure 2C a clear kinetochore signal is observed, highlighting the issues related to fixed imaging. Also, Dynein localisation during anaphase (mostly on the inward face of chromosomes) seems different from the Muscat et al., (2015) paper, where Dynein localised in the outer face of the chromosomes.

6) Does the AID allele give the same phenotype as the dhc-1(ct76ts) allele previously used by the authors or the *dhc-1(or195)*? This would also be important to make sure the effects do not depend on the GFP tag which can frequently result in a partially defective protein. These alleles have both been described as ‘fast acting’ and hence would be ideal for the comparison.

7) In Figure 2 – supplement 1 B it appears that the spindle in ‘- Auxin’ has began shortening. A proper comparison is needed using GFP::ASPM-1 in live imaging experiments.

8) There seems to be an enormous defect in segregation in Figure 2 – supplement 2. Why is this?

9) In Figure 3 how can the authors be sure they are comparing all the different conditions in Meiosis I (and not Meiosis II). In some cases, it appears that sisters, rather than homologues, are being observed. Also, the lack of consistency in the magnification for each panel does not help the reader.

10) The comparison of the monopole breakdown phenotype in Figure 3 is difficult to judge without an indication of the depletion efficiencies for lin-5 and aspm-1 RNAi. This makes the paragraph starting in line 216 highly speculative.

11) In Figure 3 – supplement 1, the lin-5(RNAi) phenotype is scarcely described in the text. For example, in monopolar spindles depleted of LIN-5 tubulin and ASPM-1 surround the chromosome surface, with Dynein exhibiting a similar localisation. Additionally, it seems again that Dynein depletion (in combination with lin-5(RNAi)) has no effect on chromosome segregation.

12) Figures 4A and B are very difficult to analyse. Also, if anything they seem to show that Dynein is not required for anaphase chromosome movement, contradicting their previous report.

13) In Figure 5 (and Figure 5 – supplement 1), SPD-1 should be analysed in live oocytes and while the analysis of SUMO localisation is informative, imaging of GFP::separase would contribute important information to the manuscript in terms of how anaphase progresses in the different conditions. This could also provide a good reference point for the unarrested videos.

14) In Figure 5A the authors claim that ‘ASPM-1 labeling could be clearly seen on both sides of these mini spindles, with some enrichment towards areas resembling spindle poles’. This is not clear to me. What is clear is the kinetochore-like signal (which the authors do not mention). Some sort of quantification for the localisation analysis should be provided. This would also benefit from live analysis to see the dynamics of ASPM-1 localisation.

15) The authors could consider depleting kinetochore Dynein recruiters to dissect the role of different populations. Depletion of RZZ components or SPDL-1 could therefore be a useful comparison in the different backgrounds.

*Reviewer #2:*

Cavin-Meza et al., investigated the role of motor proteins and their coordinated action in maintaining spindle bipolarity under acentrosomal conditions. An auxin-inducible degron system was used to rapidly remove proteins from pre-formed spindles in *C. elegans* oocytes. In addition, this approach was used to disrupt monopoles and study subsequent microtubule re-arrangement for chromosome segregation in formed miniature spindles. Specifically, the authors find that removal of dynein causes a splaying of onopolaral spindle poles. In combination with depletion of KLP-18, removal of dynein causes a disruption of monopolar spindles. This is followed by a formation of locally dispersed small spindles, which show arrangement of microtubules in between the segregated chromosomes and localization of midzone proteins, such as SPD-1. Interestingly, the authors present evidence for the kinesin-5 family motor protein BMK-1 for providing the force for this chromosome movement in these mini spindles.

The major strength of this paper is the application of the degron system to systematically manipulate oocyte spindle assembly after protein depletion. I don’t see any major weakness. This is a very interesting paper with significantly new information extending our current knowledge on acentrosomal spindles. In general, I am very positive about this manuscript. Most of the conclusions of this paper are supported by data, but some points need to be clarified and some aspects of data presentation could be worked out better.

1) In general, the text reads very well, but real numbers always have to be extracted from figures, figure legends and/or tables. The manuscript reads as a text without numbers. For each statement, the measured values, the number of measurements and the statistical data should be clearly provided also in the text. For instance, it is not clear for each presented experiment how many oocytes/samples have been analyzed.

2) The authors claim a ‚complete dynein depletion’ (in comparison to partial depletions that have been performed by others) but any quantitative information on the level of depletion is missing. Just showing LM images of oocytes with depleted GFP-dynein is not a convincing argument about a full depletion. This should be clarified.

3) I think a general control experiment is completely missing here. Have the authors ever analyzed spindles in oocytes with GFP-tubulin and without the auxin-inducible degron system after exposure to auxin? This is an absolutely necessary experiment to do. Hopefully, auxin alone does not cause splayed poles in exposed isolated oocytes. It has to be kept in mind that oocytes (without the fully developed eggshell) are extremely sensitive. Please add data on such control experiments as supplementary information.

4) There is a lot of ‚jumping’ between Figures 3 and 4 in the text (see paragraph “Dynein acts at spindle poles to promote pole formation and integrity”). The authors might re-think the order of call outs.

5) Figure 4A-B shows key information. I think the panels are way too small, and I really encourage the authors to present these interesting findings in a better way.

6) As for Figure 5 —figure supplement 1, I have a hard time to see the “onopolar persisted on what appeared to be a singular disassembling RC between each segregating chromosome pair”. A clearer image should be presented.

7) Line 384, add ‚monopolar’ to read: “resulted in a catastrophic breakdown of the onopolar spindle…”.

8) Lines 430-438, it is problematic that the authors are discussing the results of the presented manuscript in the context of unpublished data. This unpublished information should be documented somehow within this manuscript or deleted.

9) Line 479 and so forth (line 499, 529 etc.), did the authors use several ‚meiosis media’? In case only one buffer was used, change to ‚medium’. In any case, I would give the experimental conditions here and not fully rely on the details given in a reference.

10) Lines, 539-40, mention the exact auxin treatment protocols that were used within this study. Other, not applied protocols are irrelevant in the context of this paper.

The authors have to take care of these issues before this paper can be accepted for publication in *eLife*. I would be happy to review the revised version of this manuscript.

*Reviewer #3:*

Cavin-Meza et al., have parsed out the importance of different motor proteins during oocyte spindle assembly in *C. elegans*. The authors developed an auxin-inducible dynein degradation system to remove dynein from the spindle. Using this system, they addressed the question of whether dynein is needed to maintain a bipolar spindle. With the depletion of dynein after spindle assembly, the spindles form parallel arrays but still maintain chromosomes in the middle. These results suggested that dynein was involved in pole focusing. To determine further explore the role of dynein in pole focusing, they depleted KLP-18, which normally gives monopolar spindle formation. With depletion of dynein, they find that the monopoles disappear and microtubule bundles form around the chromatin. Intriguingly, the microtubule bundles can allow chromosomes to segregate at anaphase. They then address the question of what motor protein is allowing anaphase onset. They knock out BMK-1 and find that bipolar spindle assembly and chromosome segregation no longer occurs. These results show that BMK-1 contributes to spindle assembly, but somewhat redundantly with other motors. Overall, the experiments were carefully planned and executed. The development of the degradable dynein is a useful tool for the fild. The novel finding that BMK-1 does contribute to spindle assembly is important for understanding the overlapping forces on the spindle. Furthermore, the results shows that there are differences between how the motors function between mitosis and meiosis, which is intriguing and important. Only an additional control experiments is needed.

Overall, the manuscript details an important contribution of how the motor proteins are functioning together for spindle assembly. The manuscript is well-written, with clear rationales. I only have one concern:

1) The aspm-1(RNAi) and lin-5(RNAi) and strains also contain the dynein degron background with dynein tagged. Although the controls show that without auxin, dynein is clearly working, sometimes the degron tags can cause a sensitized background due to a reduction in protein levels or activity. Although the phenotypes for the single depletions have been confirmed with other studies, the aspm-1(RNAi); klp1-18(RNAi) double mutant has not. Can the authors perform the same experiment with a strain without the tagged dynein?

*Reviewer #4:*

Cavin-Meza et al., investigated the role of dynein motors in meiotic spindle pole organization and spindle function using *C. elegans* oocytes as a model. They optimized the auxin-inducible degron system to degrade endogenous AID-tagged dynein acutely, rapidly, and more completely in oocytes. They find that dynein is required for integrity of bipolar spindle poles as well as spindle monopoles induced by depletion of the kinesin-12 motor KLP-18. They further show that in the absence of dynein motors, and functional spindle poles, microtubules are still able to organize into spindle-like structures via the action of chromosomes and the kinesin-5 motor BMK-1. They also found that chromosomes can still be segregated in anaphase-like motion in these oocytes.

These results are consistent with many previous studies in mitotic and meiotic cells that investigated the function of dynein on spindle poles and chromosome segregation and showed spindle pole focusing function of this motor. Furthermore, it is known that upon removal of acentrosomal spindle pole components that assemble meiotic spindles in mouse oocytes, chromatin-mediated microtubule assembly can build spindles that can segregate chromosomes. This study is thus generally consistent with data from other models.

The authors have optimized the AID-degron system beautifully and this gives them as well as the field new opportunities to study protein function in *C. elegans* oocytes in a much cleaner and more complete depletion background. The timescale in which they can deplete endogenous proteins is impressive.

1. The authors previously published a paper where they found dynein inhibition causes chromosome segregation errors and that in this organism oocyte chromosomes are segregated independently of kinetochores in a pathway that involves dynein. While this is not in agreement with a more accepted model where inter-chromosomal microtubule arrays push chromosomes apart, it had so far provided an alternative model for oocyte chromosome segregation in the field. In this current manuscript, the authors show that dynein inhibition, presumably more complete and cleaner than other methods, in fact does not lead to any chromosome segregation errors. This is in direct conflict with their previously published work, but this major discrepancy is not addressed in the current work.

2. The authors base their hypothesis on a previously published study in human oocytes reporting that spindles fail to maintain bipolarity and mis-segregate chromosomes. This work was performed on oocytes that were unhealthy and unsuitable for IVF treatment, and it is now clear that such atretic oocytes do not represent a general oocyte form. While the maternal age effect is indeed of clinical interest, given the strong interest in many areas of cell biology in understanding the fundamental principles of chromosome segregation in oocytes (of many species), it is not necessary to base experiments on observations made using human oocytes.

3. The contribution of chromatin-mediated mitotic and meiotic spindle assembly pathways was largely ignored in the discussion although these are likely to majorly contribute to miniature spindle assembly in dynein AID oocytes. Similarly, the contribution of dynein to spindle pole focusing in other systems needs more coverage.

4. While the images are of high quality and very accessible, the lack of appropriate image analysis and quantification means the manuscript is largely descriptive. It would be important to support several conclusions made throughout the paper with robust quantification and statistical analyses.

5. While the writing is quite accessible to readers working on the *C. elegans* model, it is much less clear for those who work on other models. It would be very helpful to re-write technical bits such as ‘embryos dissected into auxin’. It is also important to include citations from other models when summarizing what is known about certain spindle features in oocytes. There is generally ambiguity in the writing that leaves the reader wondering about the precise point that is being made. E.g. microtubule bundles were able to remain relatively aligned’ – this and several other statements in the paper are very descriptive and without quantitative image analysis to back them up, it is difficult to grasp what they mean.

This manuscript can be significantly improved by providing a detailed discussion concerning the conflict between the data within and the lab's previously published work that shows dynein inhibition causes chromosome segregation errors. Furthermore, robust image quantification and statistical analyses are needed to make the paper much less descriptive cell biology than it is now. It would also be good to include some controls to show that 4 hour treatment of non-AID strain wild-type oocytes has no effect on several aspects of spindle assembly and function studied here.

---

## [Author Response]

Essential revisions:While all reviewers found your work very interesting, there was agreement that a number of important issues need to be addressed in a revised version. The following is a summary of the main concerns that have to be addressed and you will find the detailed reviews below.1) The phenotypes should be fully analysed and described quantitatively as much as possible. For each statement, the measured values, the number of measurements and the statistical data should be clearly provided also in the text.

To better support our conclusions, we have made efforts to provide more rigorous quantification of numerous aspects of our data. First, we performed new pole splaying quantifications. Instead of judging by eye whether a pole was “focused” or “splayed” (as we did in the original submission), we measured the widths of the poles and of the center of the spindle; we then calculated a ratio of the pole to equator width as a measurement of how splayed the poles are (this is shown in Figure 2B). This new analysis confirmed the finding that we reported in our original manuscript – that the poles are substantially splayed upon auxin treatment. Additionally, we have added rigorous controls to demonstrate that the effects we see are not caused by auxin treatment (Figure 2 —figure supplement 2). We dissected oocytes lacking degron-tagged dynein into auxin and then imaged spindles over time, demonstrating that neither the spindle length nor the pole-to-equator ratio changed. Finally, we now report the number of images analyzed for all figures, we have added quantitative information into the manuscript when discussing results, and we have tried to emphasize the more quantitative aspects of our paper in our revised manuscript, such as our measurements of chromosome segregation rates and distance traces in live imaging.

2) The newly generated AID strain(s) need to be more thoroughly characterised:a) Quantitative information on dynein depletion and its comparison with previously used methods (RNAi and ts alleles) is needed;b) it would also be important to make sure none of the observed effects depend on the AID or AID::GFP tag which can frequently result in a partially defective protein.c) control that auxin alone (without any AID) does not cause splayed poles in exposed isolated oocytes.

a) In the original version of the manuscript we made a few statements implying that we were achieving “complete” dynein depletion using our AID method, contrasting this to previous studies by ourselves and others that used partial RNAi or temperature-sensitive mutants. However, the reviewers rightly pointed out that since we had not quantified the level of depletion, we could not make this statement. Therefore, for the revision, we attempted to quantitatively assess dynein depletion using western blotting. Unfortunately, we had difficulties transferring DHC-1, given the large size of this protein (especially with the GFP and degron tags added); we extensively tried to optimize this DHC-1 western but we were unsuccessful, despite getting westerns to work for other AID strains in our lab (for one example, see our recent BioRxiv preprint, Mullen and Cavin-Meza, *et.al.*, 2022). We therefore ask that this experiment not be required for acceptance of the manuscript, as the data we present is consistent with substantial dynein depletion. We show a drastic reduction of DHC-1::GFP signal upon dissection of oocytes into auxin, undetectable dynein via immunofluorescence following auxin treatment, and the phenotypes that we report are highly reproducible, suggesting that there is not a lot of variability in the efficiency of protein depletion between experiments. However, since we recognize that we have been unable to demonstrate that our depletion is in fact complete (or to compare the level of depletion to the other methods, due to the problems with western blotting noted above), we have revised our manuscript to soften our language, and we no longer claim that we have achieved complete depletion. The revised wording therefore more accurately describes our results.

b) The phenotypes observed in our dynein depletion conditions do not appear to depend on the GFP tag. The pole splaying phenotypes that we observe following dynein depletion are very similar to what has been reported for partial dynein or dynactin depletion via RNAi by another lab, so pole splaying on bipolar spindles does not appear to depend on the GFP tag. Moreover, we previously reported the results of partial dynein inhibition on monopolar spindles (Muscat, *et.al.* 2015); our goal when we performed those original experiments was to ask whether dynein provided chromosomes with a minus-end directed force (i.e., would chromosomes be found further out towards the plus ends on a monopolar spindle following dynein depletion?). When we optimized those experiments we tried multiple conditions for partial dynein RNAi (i.e., different amounts of time on the RNAi plates) and we noticed that with longer incubations on the RNAi plates, we could not find any monopolar spindles – chromosomes appeared dispersed throughout the oocyte (see Author response image 1).

**Author response image 1. sa2fig1:** 

At the time we did not fully understand this phenotype, so we only analyzed oocytes where monopolar spindles were intact (with more partial dynein depletion conditions) to address our question about forces on chromosomes. However, given the results presented in the current manuscript, we now understand that we could not find monopolar spindles with more complete dynein depletion because the monopole had blown apart, just like what we report in the current manuscript using the degron strain. This past observation suggests that the monopole dissolution phenotype does not depend on the GFP tag either.To speak to the functionality of DHC-1::degron::GFP, we have never observed defects in any aspect of the meiotic divisions in the Dynein AID strain in the absence of auxin. In the original Zhang *et al.,* paper from the Dernburg lab, where this strain was first reported, no significant changes in brood size or percentage of male progeny were detected (this information is found in their supplemental tables); we have now added this information to our Results section. In our own hands, we similarly have not observed anything in worm upkeep (development, viability, brood size, etc.) that would suggest that there are unintended consequences of tagging DHC-1, and we have not perceived any changes in spindle morphology in the absence of auxin in this strain, when compared to wild-type spindles (see various control images and videos throughout our manuscript). Therefore, while we cannot be 100% certain that the tagged DHC-1 is fully wild-type, we don’t think that there are major issues with this strain that would affect the conclusions of our manuscript.

c) We have now performed this important control experiment utilizing a worm strain expressing TIR1 and same fluorophores (GFP::tubulin and GFP::histone), but lacking the degron-tagged version of DHC-1 (this experiment is now presented in Figure 2 —figure supplement 2). In this strain, exposure of isolated oocytes to auxin has no noticeable effect on spindle morphology; spindle length remained constant and the ratio between the pole and equator widths (a measurement of pole splaying, as described in response to point #1) also did not change.

3) Assessment of protein depletions should be presented, more so in cases where a 'negative' result is shown (see the examples mentioned in the review reports regarding lin-5 and aspm-1).

In the original manuscript, we presented double RNAi of *klp-18(RNAi)* with either *lin-5(RNAi)* or *aspm-1(RNAi)*, and we observed that some monopolar spindles broke down. This phenotype was similar to the dynein depletion phenotype on monopolar spindles, but was less penetrant since some monopoles were intact; this is the negative result noted above by the reviewer. In the revision, we re-evaluated the inclusion of this partial breakdown result, given the difficulties in interpreting these data. Therefore, this negative result is no longer presented in the revised manuscript. For an explanation of why we have not added quantification of dynein depletion, please see point #2 above.

4) Data in the present manuscript, as it stands, shows that the lab's previous suggested model of Dynein action on chromosome movement is compromised. Also, Dynein localisation itself is different in some cases when compared to the Muscat et al., (2015) paper. Without proper discussion of this issue including plausible explanations for this discrepancy, the reader is left wondering what the actual functions of Dynein are.

It is true that we previously hypothesized in Muscat *et al.,* (2015) that dynein contributed to chromosome segregation; this was based on evidence that (1) chromosomes are subjected to minus-end forces prior to anaphase, (2) dynein was observed near the outside surface of segregating chromosomes, and (3) there were lagging anaphase chromosomes in a dynein temperature-sensitive mutant following a shift to the restrictive temperature. However, in that paper we noted that the dynein ts mutant phenotype was not completely penetrant – although we saw lagging chromosomes, chromosome segregation was not completely blocked. Therefore, in the discussion we were open about the fact that, even if our model that dynein facilitated chromosome segregation was correct, that other mechanisms may contribute. In particular, we stated that since the spindle elongates in anaphase, that it was “likely that spindle elongation also contributes to chromosome segregation”. In the years since our paper was published, it has become clear based on the work of multiple labs that this anaphase B-like spindle elongation is the major driver of chromosome segregation, and that, if dynein does play a role in segregation, it is redundant with other mechanisms. However, since we never held the viewpoint that dynein was the only driver of chromosome segregation, we do not see a major contradiction between our 2015 study and the conclusions of the current study. The findings in our 2015 paper and current manuscript are also consistent; dynein depletion does cause lagging chromosomes (seen in our original study and also in Figure 2 —figure supplement 3 of the current study), and dynein does localize near the outer surface of chromosomes (seen in our original study and also in Figure 1C of the current study, although it is now recognized based on the work of other labs that this likely represents pole, rather than chromosomal staining). Therefore, we simply see this as an example of the field moving forward in the 7 years since our first study was published, changing the interpretation of some of our original experiments. Our new study reinforces the current view that chromosome segregation does not require dynein, by revealing another experimental condition where chromosomes can segregate following dynein depletion (in this case also in the absence of KLP-18); chromosome segregation under these conditions is likely driven by anaphase B-like spindle elongation, since the miniature spindles lengthen as chromosomes segregate. We have now included a discussion of these issues related to our 2015 paper in our Discussion section.

5) Results should be based on live imaging experiments whenever possible and the quality of the videos needs to be improved. For example, Figure 2 – supplement 2 is a key Figure and these videos should start with properly assembled Metaphase I spindles and from there, pole disassembly should be assessed. Also, as presented, the difference in intensity between – and + auxin makes this result not entirely convincing to me.

In our manuscript, all of the major conclusions are supported by both live and fixed imaging. Live imaging is used to show that (1) acentrosomal spindle poles splay following acute dynein depletion, (2) monopoles break apart following dynein depletion, (3) miniature spindles form that support chromosome segregation following monopole breakdown, and (4) that the formation of these miniature spindles requires BMK-1. We think that the live and fixed imaging methods used in the manuscript are complementary, as live imaging provides essential dynamic information, while fixed imaging provides higher resolution images and the ability to easily assess multiple spindle markers, to better understand the phenotypes.

In the revision, we have made efforts to generate additional live imaging data and to improve the quality of our videos as requested. In the original submission, we had a few videos, such as the ones mentioned in Figure 2 —figure supplement 2 (now Figure 2 —figure supplement 3), where the intensity of the GFP::histone signal was different when comparing the control (auxin) and experimental (+auxin) conditions. We do acknowledge that we frequently observe some variation in fluorescence intensity between videos. This is because we utilize the same laser power, EM Gain, and exposure timings across every video for a given worm strain. However, the embryo is a three-dimensional object, and the depth of the spindle in the Z-plane (i.e., how close the spindle is to the coverslip) can significantly impact the clarity and brightness of the timelapse. We do not think that these differences in signal intensity are indicative of atypical variance in expression levels, and we also do not think that they reflect poorly on the quality of our data, as the phenotypes that we observe are very consistent, despite the differences in the GFP signal. However, to make the videos easier to evaluate, we have generated new videos, and we have swapped out the timelapses highlighted in our figures with better matched control and experimental examples (i.e., where the intensity of the fluorophores are more similar between the two conditions). The phenotypes observed in the new videos are identical to those reported in the original version of the manuscript.

We also agree with the reviewer that we would ideally start the timelapses in Figure 2 —figure supplement 2 (now Figure 2 —figure supplement 3) with fully assembled Metaphase I spindles, to show that the poles of pre-formed unarrested spindles splay upon dynein depletion (as we show in our metaphase-arrest experiments). However, when we start with unarrested Metaphase I spindles and begin filming, these spindles have usually shortened and transitioned into anaphase before dynein is fully depleted, complicating analysis. Therefore, we instead scan the slide for embryos where the first polar body has just been extruded (where the Meiosis II spindle has not yet formed) and we immediately start filming. In these videos, spindles begin to form as dynein is being depleted, by the time these spindles reach metaphase, the poles are clearly splayed. Although this is not a perfect experiment, it is the best we can do, given how long unarrested oocytes typically stay in metaphase (not long), and the timing of dynein depletion.

We think that these improvements to the manuscript better support our conclusions, and hope that they will be acceptable to the reviewers.

6) Unpublished data should be documented within this manuscript or not discussed at all.

In our discussion, we have a few sentences describing results from another study performed in our lab; when we submitted our original manuscript this other study was unpublished. However, we recently uploaded this work to BioRxiv (Mullen and Cavin-Meza, *et al.,* 2022), so we now cite this preprint in the current manuscript.

Reviewer #1:

Recommendations for the authors

1) The authors have an extremely useful system to analyse disassembly but need to make full use of it. They need make sure they image the newly fertilised oocyte (first one after the spermatheca), therefore avoiding side effects due to prolonged arrest.

We agree that avoiding embryos that have been arrested for long periods of time is critical. However, it is difficult to implement this specific suggestion of the reviewer for technical reasons. For our two major assays (live imaging and fixed immunofluorescence), we need to dissect the oocytes out of the worm to image them, so we lose the positional information of the germ line and we cannot tell which oocyte is the most recently fertilized. Although in principle we could soak whole worms in auxin and then image depletion of dynein in an intact worm (so we can make sure to image the most newly fertilized oocyte, as the reviewer suggests), we have found that it takes substantially longer to deplete proteins when the oocytes are not dissected, likely because it takes additional time for the auxin to reach the oocytes (we discussed these various methods in more detail in Divekar, et.al. 2021, *Current Protocols*). Therefore, it would not be possible to do the type of rapid depletion experiments that we present in the paper (depleting dynein within minutes), in a situation where we could retain the positional information of the germ line.

However, after looking at many hundreds of metaphase-arrested embryos in both live and fixed imaging conditions, we are confident that we can accurately identify the hallmarks of prolonged arrest (reduced dynamics of spindle and chromosomes, stretching of chromosomes, gradual widening of spindle midzone). For the experiments in the manuscript, we did not analyze any oocytes that displayed these phenotypes, which we hope minimizes this concern of the reviewer. We have now added a sentence to the Materials and methods (section on live imaging) to make it clear that we excluded some oocytes from imaging for this reason. In addition, for the major experiments of the paper, we also show similar phenotypes in unarrested conditions, so we do not believe that the metaphase arrest influences the conclusions of our manuscript.

2) It is not clear how the authors define t = 0. This is not a trivial issue, and it is vital to know whether the two oocytes being analysed have the same 'history' (i.e. been arrested for ~ same time). For example, if one oocyte has just undergone fertilisation whereas the other has been arrested for longer, susceptibility to Dynein depletion could be drastically altered.

This is a good point, and we have now described how we define t=0 in in the manuscript, in the “*Ex utero* live imaging” section of the Materials and methods. For embryos that are not arrested, we have acquired a set of new videos where t=0 is the onset of Meiosis II. We believe that this is a reasonable point for synchronization because Metaphase I is difficult to catch immediately after dissection (it requires considerable luck to find an embryo at metaphase I shortly after mounting the slide). Additionally, when we catch Metaphase I spindles that are unarrested, by the time DHC-1 is depleted from embryos, these spindles have usually shortened and transitioned into anaphase, complicating analysis. Therefore, we changed our strategy to instead scan the slide for embryos where the first polar body has just been extruded (where the meiosis II spindle has not yet formed); this is t=0.

For our metaphase-arrested videos, t=0 is defined as the onset of filming (roughly 5 minutes post-dissection, depending on the speed of mounting the slide and how quickly we are able to find a suitable embryo). We acknowledge that it is possible that the duration of arrest could have unforeseen consequences on depletion phenotypes. However, we exclude embryos that appear to be exhibiting prolonged metaphase arrest (see point #1 above), and the phenotypes from the embryos that we observe are extremely consistent despite the lack of perfect synchronization -- particularly the pole splaying phenotypes we observe in both arrested and unarrested conditions (i.e., Figure 2D versus Figure 2 —figure supplement 3). Therefore, we do not think that the duration of the arrest, if it indeed has an effect on the susceptibility to dynein depletion, is having a big enough effect to alter our conclusions; if this were the case then we think that we would see more variability in our phenotypes.

3) The lack of consistency in the signal intensity is concerning (see Figure 2D, GFP::histone signal differs massively between '- Auxin' and '+ Auxin').

We agree that there were differences in the intensity of the GFP::histone signal between the -Auxin and +Auxin conditions of Figure 2D in the original manuscript; to address this concern we have replaced the original -Auxin video with a newly acquired one where the GFP::histone signal is more similar to the +Auxin video. We do acknowledge that we frequently observe some variation in fluorescence intensity between videos. This is because we utilize the same laser power, EM Gain, and exposure timings across every video for a given worm strain. However, the embryo is a three-dimensional object, and the depth of the spindle in the Z-plane (i.e., how close the spindle is to the coverslip) can significantly impact the clarity and brightness of the timelapse. We do not think that these differences in signal intensity are indicative of atypical variance in expression levels, and we also do not think that they reflect poorly on the quality of our data, as the phenotypes we observe are very consistent, despite the differences in the GFP signal.

4) In Figure 2 – supplement 2, the issue highlighted in 3) is even more concerning, but additionally, and equally or more important, the authors should be able to start the imaging experiment from a 'normal' metaphase in both conditions. Without this careful comparison (in addition to imaging GFP::ASPM-1 as described below) the claims about the pole stability defects are not entirely clear.

We agree with this point. We have now taken additional videos of unarrested embryos so that the videos in Figure 2 – supplement 2 (now Figure 2 – supplement 3) have been replaced; we also now show two videos for both the +Auxin and -Auxin condition. For these new videos we synchronize t=0 to the onset of Meiosis II (please revisit point #2 for an explanation of why we chose this timepoint, and why we cannot start imaging at metaphase, as suggested by the reviewer). See point #7 below for in-depth discussion of GFP::ASPM-1 imaging.

5) Why is there no Dynein signal on kinetochores as reported previously? Is this related to fixation? In fact, In Figure 2C a clear kinetochore signal is observed, highlighting the issues related to fixed imaging. Also, Dynein localisation during anaphase (mostly on the inward face of chromosomes) seems different from the Muscat et al., (2015) paper, where Dynein localised in the outer face of the chromosomes.

It is true that we see kinetochore staining more clearly in live imaging with the mCherry::Tub, DHC-1::deg::GFP strain. However, we do often see kinetochore staining in fixed imaging as well. Kinetochore staining is present in many of our images (see Figure 1C, metaphase, Figure 1E, -Auxin control; you can see faint cup-like staining adjacent to the bivalents); although we realize that this is hard to see in the projection image in the figure, when we look through z-stacks the kinetochore staining is more obvious in single slice images. Therefore, we think that dynein on the spindle obscures the kinetochore pattern, making it less obvious in fixed images of bipolar spindles. Supporting this interpretation, kinetochore staining is much clearer when spindle-associated dynein is reduced (see Figure 1 —figure supplement 2B; *aspm-1(RNAi)* image) or when the chromosomes are more spread out, reducing interference from spindle-localized dynein (see the various fixed monopolar spindle images throughout the manuscript). Therefore, we do not think that our data conflicts with published work. In terms of anaphase, while we sometimes observe some DHC-1 staining near the inside face of chromosomes, we do not consider this stronger than the localization observed on the poleward side of chromosomes (very common across all of our images).

In addition to this specific point, we would also like to address the concerns raised about our fixed imaging experiments more generally. While we certainly agree that fixed imaging has caveats (no tracking of dynamics, potential artifacts from fixation, non-specific binding of antibodies), we also wish to emphasize that fixed imaging can have significant benefits over live imaging (improved resolution of signal, ability to visualize numerous channels simultaneously, can obtain data quickly without months of strain generation and crosses that be genetically difficult). We believe that both fixed and live imaging have specific contexts in which they are the better-suited method of acquisition. Therefore, we have strived to utilize both methods in our study and we think that these imaging modalities are complementary; although there may be some slight differences in our data depending on the method by which they were acquired (e.g. the different levels of dynein detected at kinetochores, as noted by the reviewer), we don’t think that any of the major conclusions of our manuscript are compromised by these caveats.

6) Does the AID allele give the same phenotype as the dhc-1(ct76ts) allele previously used by the authors or the dhc-1(or195)? This would also be important to make sure the effects do not depend on the GFP tag which can frequently result in a partially defective protein. These alleles have both been described as ‘fast acting’ and hence would be ideal for the comparison.

We have not performed direct comparisons between our Dynein AID strain and either of the mentioned fast-acting *ts* alleles; since we do not have a microscope stage that can be quickly heated, assessing whether a shift to the restrictive temperature affects the organization of preformed spindles would be technically challenging for us with our available equipment. Therefore, we respectfully ask that these experiments not be required for acceptance.

Regardless, we do not think that the phenotypes seen in our dynein depletion conditions depend on the GFP tag. The pole splaying phenotypes that we observe following dynein depletion are very similar to what has been reported for partial dynein or dynactin depletion via RNAi by the McNally lab, so pole splaying on bipolar spindles does not appear to depend on the GFP tag. Moreover, we previously reported the results of partial dynein inhibition on monopolar spindles (Muscat, et.al. 2015); our goal when we performed those original experiments was to ask whether dynein provided chromosomes with a minus-end directed force (i.e., would chromosomes be found further out towards the plus ends on a monopolar spindle following dynein depletion?). When we optimized those experiments we tried multiple conditions for partial dynein RNAi (i.e., different amounts of time on the RNAi plates) and we noticed that with longer incubations on the RNAi plates, we often could not find any monopolar spindles – chromosomes appeared dispersed throughout the oocyte with associated microtubules (see Author response image 2). At the time we did not fully understand that phenotype, so we only analyzed oocytes where monopolar spindles were intact (with more partial dynein depletion) to address our question about forces on chromosomes. However, given the results presented in the current manuscript, we now understand that we could not find monopolar spindles with more complete dynein depletion because the monopole had blown apart, just like what we report in the current manuscript using the degron strain. This observation using RNAi suggests that the monopole dissolution phenotype does not depends on the GFP tag either. Since we made these observations during experimental optimization, we did not quantify the phenotype at the time and so we are not able to add these data to the manuscript without re-doing those partial depletion experiments to generate more n’s. Therefore, in the interest of time, we ask that this not be required for acceptance (though if the reviewer considers this an essential experiment, we certainly could do it again).

To speak to the functionality of DHC-1::degron::GFP, we have never observed defects in any aspect of the meiotic divisions in the Dynein AID strain (CA1215) in the absence of auxin. To our eyes, all of our control experiments with this strain present normally when compared to spindles from other common lab strains such as N2, EU1067 (GFP::tubulin; GFP::histone), or OD868 (GFP::tubulin; mCherry::histone). While we understand that adding any tag to a protein has the possibility of interfering with functionality, we have not observed anything in worm upkeep (development, viability, brood size, etc.) or experimentally that would suggest that there are unintended consequences of tagging DHC-1. In the original Zhang et al., paper from the Dernburg lab, where this strain was first reported, no significant changes in brood size or % of male progeny were detected (found in their supplemental tables); we have now added this information to our Results section and we have cited their paper in the relevant section of the manuscript.

7) In Figure 2 – supplement 1 B it appears that the spindle in '- Auxin' has began shortening. A proper comparison is needed using GFP::ASPM-1 in live imaging experiments.

We went back to the original image file for this particular control spindle, and we noticed that it is slightly tilted in the z-plane, which we think gave the appearance that it is shortening. However, the measured pole to pole length of the image in question is 6.13 microns, very close to the mean spindle length calculated in our data (6.07 microns). If this is truly a point of contention, we will happily remove this particular image and replace it with a different representative image.

We agree that imaging ASPM-1 live would complement the fixed imaging analysis in Figure 2 – supplement 1, and in principle this is a good idea. However, since it would require a substantial amount of work (due to some issues with strain construction), we are of the opinion that it may not be worth the effort. In principle, we could cross an existing GFP::ASPM-1 strain into the Dynein-AID strain. However, since Dynein is also tagged with GFP in this strain, we believe that this will make it difficult to interpret GFP::ASPM-1 localization during our videos. Since DHC-1 is on poles and the spindle itself, it would be hard to determine what signal was truly ASPM-1 until DHC-1 was fully depleted, which somewhat defeats the purpose of filming ASPM-1 localization in the first place. An alternative would be to use CRISPR/Cas9 to tag ASPM-1 with mCherry in our Dynein AID strain, or vice versa (DHC-1::degron::mCherry into EU2876, the GFP::ASPM-1 strain). While this is a feasible experimental setup, we do not believe that the work and time investment of generating either of these strains would be worth it. The data that we present in our linescan analysis of fixed images is consistent and reproducible, and the pole splaying phenotype that we observe in both live and fixed imaging is extremely clear (with and without being able to see ASPM-1 localization).

While we do agree that observing ASPM-1 using live imaging could further support some of our observations, we anticipate that we might also lose some of the resolution that we took advantage of for this linescan quantification (please note the tightness of our averages and SEM error bars). Therefore, due to the complications (and time investment) of generating a new strain using CRISPR, and our perception that it would not significantly improve the quality of our data, we have opted to not perform this experiment for this revision and we respectfully ask that it not be required for acceptance of our manuscript.

8) There seems to be an enormous defect in segregation in Figure 2 – supplement 2. Why is this?

This is correct; in our original representative video for this supplement, the spindle in question had failed to extrude the first polar body in Meiosis I, resulting in additional chromosomes being present during Meiosis II. We think that this is one cause of the lagging chromosomes in anaphase. Moreover, this phenotype is also consistent with previous studies that reported lagging chromosomes following dynein depletion in oocytes (one example is our own previous work in Muscat, et.al. 2015, and this same observation was subsequently made by McNally et.al. 2016). Although we originally thought that this would be acceptable to use as the representative video even though the polar body had failed to extrude (since it showed the pole splaying phenotype), after considering point #2 above about synchronization of videos to a standardized t=0, we have replaced this +Auxin video with two newly acquired videos, both synchronized to start filming at the onset of Meiosis II (with clear evidence of a successful polar body extrusion in Meiosis I).

9) In Figure 3 how can the authors be sure they are comparing all the different conditions in Meiosis I (and not Meiosis II). In some cases, it appears that sisters, rather than homologues, are being observed. Also, the lack of consistency in the magnification for each panel does not help the reader.

It is true that our original Figure 3 contained examples of both Meiosis I and Meiosis II spindles. We found that depletion of dynein from either MI or MII monopolar spindles caused the monopoles to blow apart; there was no difference in the phenotype, so we showed examples of both. While the chromosomes are different between Meiosis I and II (bivalents vs. individual chromosomes), we are not aware of any literature that has demonstrated major differences in the organization of acentrosomal spindles when comparing the two meiotic divisions in *C. elegans*. Therefore, we do not see any inherent complication in assessing both MI and MII spindles and showing examples of both (since the phenotypes that we report are the same for both divisions). However, if there is a reason we have not thought of for why we should not treat these spindles as equal, we would appreciate that information, so that we can reconsider this point and adjust this figure.

Regarding the differences in magnification, we agree that it would in theory be better to crop everything the same for consistency. However, we cropped images the way we did to maximize the clarity of each image. Specifically, in cases where the images have been considerably zoomed out, we felt that this better captured the sense of monopole breakdown in relation to the distance across the cytoplasm; the reader can get a better sense of just how far these chromosomes have drifted apart when we present a more zoomed out image. We understand that this creates inconsistency in the figure, since the control monopolar spindles are zoomed in more. However, if we cropped these control images so that they were zoomed out a similar amount, the spindle would be much smaller (and harder to see), so we would prefer not to do this. Instead, we put a sentence in the Figure 3 legend, explaining to the reader that the +Auxin images are zoomed out more (to make sure they notice this when evaluating our images). However, if the reviewer feels strongly about this point, we would gladly reconsider redoing our cropping.

10) The comparison of the monopole breakdown phenotype in Figure 3 is difficult to judge without an indication of the depletion efficiencies for lin-5 and aspm-1 RNAi. This makes the paragraph starting in line 216 highly speculative.

This is a fair criticism and we have re-evaluated the inclusion of our partial breakdown of monopolar spindles using either *lin-5* or *aspm-1* RNAi, given the difficulties in interpreting these data. Upon reflection, we have decided to restructure our *lin-5* and *aspm-1* RNAi data, removing the data in question (See Figure 1 —figure supplement 2), and we have adjusted the manuscript accordingly.

11) In Figure 3 – supplement 1, the lin-5(RNAi) phenotype is scarcely described in the text. For example, in monopolar spindles depleted of LIN-5 tubulin and ASPM-1 surround the chromosome surface, with Dynein exhibiting a similar localisation. Additionally, it seems again that Dynein depletion (in combination with lin-5(RNAi)) has no effect on chromosome segregation.

As noted in response to point #10, we have restructured our *lin-5* and *aspm-1* RNAi data and have adjusted the manuscript to discuss our findings in a different manner (Please see Figure 1 —figure supplement 2 and corresponding text in the Results section describing this figure). This reorganization has removed the data highlighted in this comment. We have discussed dynein depletion and chromosome segregation in the next point (#12).

12) Figures 4A and B are very difficult to analyse. Also, if anything they seem to show that Dynein is not required for anaphase chromosome movement, contradicting their previous report.

In response to this and a similar comment from Reviewer 2 (point #5), we have reorganized Figure 4 so that we could make the images larger and easier to see; we also added zooms that highlight the key findings – that microtubules are able to reorganize and chromosomes are able to segregate on these miniature spindles. We hope that these improvements will make the data easier to evaluate.

To speak to the comment of how this data may contradict our previous 2015 paper in *eLife* (also raised by reviewer 3, point #1), the reviewer is correct that we previously hypothesized that dynein contributed to chromosome segregation; this was based on evidence that chromosomes are subjected to minus-end forces prior to anaphase, and our observation that there were lagging chromosomes in a dynein temperature sensitive mutant following a shift to the restrictive temperature. However, in that paper we noted that the phenotype was not completely penetrant – although we saw lagging chromosomes, chromosome segregation was not completely blocked. Therefore, in the discussion we were open about the fact that, even if our model that dynein facilitated chromosome segregation was correct, that other mechanisms may contribute. In particular, we stated that since the spindle elongates in anaphase, that it was “likely that spindle elongation also contributes to chromosome segregation”. In the years since our paper was published, it has become clear based on the work of multiple labs that this anaphase-B like spindle elongation is the major driver of chromosome segregation, and that, if dynein does play a role in segregation, it is redundant with other mechanisms. However, since we never held the viewpoint that dynein was the only driver of chromosome segregation, we do not see a major contradiction between our 2015 study and the current study. The findings in our 2015 paper and current manuscript are also consistent; dynein depletion does cause lagging chromosomes (seen in our original study and also in Figure 2 —figure supplement 3 of the current study). Therefore, we simply see this as an example of the field moving forward in the 7 years since our first study was published, changing the interpretation of our original experiments. Our current study reinforces this updated view, showing another experimental condition where chromosomes can segregate following dynein depletion (in this case also in the absence of KLP-18). We have now included a more thorough discussion of this issue in our Discussion section.

13) In Figure 5 (and Figure 5 – supplement 1), SPD-1 should be analysed in live oocytes and while the analysis of SUMO localisation is informative, imaging of GFP::separase would contribute important information to the manuscript in terms of how anaphase progresses in the different conditions. This could also provide a good reference point for the unarrested videos.

Although in principle it would be nice to have live imaging of SPD-1 to complement our fixed imaging, we do not believe that the perceived benefits of utilizing live instead of fixed imaging are large enough to warrant the time it would take to do this experiment. SPD-1 localization is only apparent during a short window of time (anaphase spindles that exist for ~5 minutes), and we think that it would be hard to catch this window (and to interpret the resulting videos) if the chromosomes were not also labeled in this strain. Therefore, we would have to cross both SPD-1::GFP and mCherry::histone into our Dynein degron strain, but the main mCherry::histone transgene that people use (made by the Desai lab in strain OD56), is on chromosome IV, the same chromosome as TIR1. Therefore, this would be a very complicated cross and it require a lot of effort to generate the necessary strain (and then of course there would be additional time to acquire the actual videos). Since we do not think that our fixed imaging is missing any crucial localization information, we don’t think that this suggested experiment would be worth potentially months of work. As an alternative, we have repeated our fixed SPD-1 staining again (imaged an additional ~50 embryos in auxin treatment) to strengthen our SPD-1 data, and we have replaced the original representative images with clearer examples; we hope that this is satisfactory to the reviewer.

We greatly appreciate the suggestion of imaging separase. Due to the difficulties in performing live imaging (which would involve the same strain construction issues noted above, since we would need to use mCherry::histone to label chromosomes), we have chosen to perform fixed imaging to assess separase. We have generated a brand new supplemental figure (Figure 5 —figure supplement 2) to demonstrate that separase localization on miniature anaphases is consistent with what has been seen on wild-type anaphase spindles.

14) In Figure 5A the authors claim that 'ASPM-1 labeling could be clearly seen on both sides of these mini spindles, with some enrichment towards areas resembling spindle poles'. This is not clear to me. What is clear is the kinetochore-like signal (which the authors do not mention). Some sort of quantification for the localisation analysis should be provided. This would also benefit from live analysis to see the dynamics of ASPM-1 localisation.

We have updated our manuscript to mention the off-target kinetochore staining that sometimes occurs when utilizing this ASPM-1 antibody (originally documented in Wignall and Villeneuve 2009); we had originally pointed out this background staining only in the figure legend, but now we have also added this information to the Results section so that it will be easier for the reader to find. While this image unfortunately does have more ASPM-1 background staining than most of our other images, we believe that our observation of ASPM-1 signal on either side of these individual, seemingly bipolar spindles around individual chromosomes is real and indicative of some form of microtubule reorganization occurring following monopole breakdown (singular polarity) to mini anaphases (bipolarity). To address the reviewer’s comment, we have softened our language describing this data, removing the word “clearly” from the quoted sentence and changing the text from “some enrichment” to “minor enrichment” when referring to the localization of ASPM-1 to pole-like structures. For our thoughts on assessing GFP::ASPM-1 localization, please revisit point #7.

15) The authors could consider depleting kinetochore Dynein recruiters to dissect the role of different populations. Depletion of RZZ components or SPDL-1 could therefore be a useful comparison in the different backgrounds.

This a good suggestion. However, we are not sure that performing these experiments would provide substantial new insights since two other publications have already examined the role of kinetochore-localized dynein for chromosome segregation (Laband et al., 2017 and Danlasky et al., 2020). Between the two studies, they depleted ROD-1, ZWL-1, and SPDL-1; none of these depletions yielded a change in chromosome segregation rate/distance during anaphase. Therefore, we are not sure what new information we would learn if we performed similar experiments. Notably, Danlasky et al., also utilized Dynein AID and noted that Anaphase B velocity increased from wild-type anaphase spindles. This finding is in strong agreement with our chromosome segregation traces (Figure 4C) and suggests that dynein activity on the spindle, not at kinetochores, has an effect on anaphase spindle elongation.

Reviewer #2:Cavin-Meza et al., investigated the role of motor proteins and their coordinated action in maintaining spindle bipolarity under acentrosomal conditions. An auxin-inducible degron system was used to rapidly remove proteins from pre-formed spindles in *C. elegans* oocytes. In addition, this approach was used to disrupt monopoles and study subsequent microtubule re-arrangement for chromosome segregation in formed miniature spindles. Specifically, the authors find that removal of dynein causes a splaying of onopolaral spindle poles. In combination with depletion of KLP-18, removal of dynein causes a disruption of monopolar spindles. This is followed by a formation of locally dispersed small spindles, which show arrangement of microtubules in between the segregated chromosomes and localization of midzone proteins, such as SPD-1. Interestingly, the authors present evidence for the kinesin-5 family motor protein BMK-1 for providing the force for this chromosome movement in these mini spindles.The major strength of this paper is the application of the degron system to systematically manipulate oocyte spindle assembly after protein depletion. I don’t see any major weakness. This is a very interesting paper with significantly new information extending our current knowledge on acentrosomal spindles. In general, I am very positive about this manuscript. Most of the conclusions of this paper are supported by data, but some points need to be clarified and some aspects of data presentation could be worked out better.1) In general, the text reads very well, but real numbers always have to be extracted from figures, figure legends and/or tables. The manuscript reads as a text without numbers. For each statement, the measured values, the number of measurements and the statistical data should be clearly provided also in the text. For instance, it is not clear for each presented experiment how many oocytes/samples have been analyzed.

This is a good suggestion and we have updated numerous areas in the Results section to address this point.

2) The authors claim a ‚complete dynein depletion’ (in comparison to partial depletions that have been performed by others) but any quantitative information on the level of depletion is missing. Just showing LM images of oocytes with depleted GFP-dynein is not a convincing argument about a full depletion. This should be clarified.

We completely agree with the reviewer that we would ideally present stronger evidence for the conclusion that we are completely depleting dynein. We therefore tried to perform a western blot to more quantitatively assess the level of DHC-1 depletion. However, we were unsuccessful in our attempts, despite being able to successfully perform westerns on other degron-GFP-tagged strains. We think that this is because dynein is very large when tagged (around 500kD) and this makes it difficult to transfer. Therefore, since we were not able to do this experiment, we have removed all sentences from the manuscript that previously stated that we were completely depleting dynein, to soften our conclusions. We now mostly emphasize that the auxin system is powerful because it allows rapid depletion (rather than emphasizing that it allows “complete” depletion). The only remaining sentence that claims that our depletion may be better than that achieved in previous studies is in the first paragraph of the Results section, where we note that most previous studies used partial RNAi and we state that we are “attempting” to achieve “more complete” depletion. We think that this is fair, since past studies using partial RNAi have intentionally scaled back the amount of depletion (to avoid developmental defects), and we do not have to do that with the auxin system; therefore, it is likely that our depletion is more complete (even if we cannot show, and therefore don’t state, that it is fully complete). We hope this addresses the reviewer’s concern.

3) I think a general control experiment is completely missing here. Have the authors ever analyzed spindles in oocytes with GFP-tubulin and without the auxin-inducible degron system after exposure to auxin? This is an absolutely necessary experiment to do. Hopefully, auxin alone does not cause splayed poles in exposed isolated oocytes. It has to be kept in mind that oocytes (without the fully developed eggshell) are extremely sensitive. Please add data on such control experiments as supplementary information.

We agree that this is a crucial control. When we first started using the degron system in our lab, we performed multiple controls and we generated concrete data showing that there are no side effects of auxin treatment. However, we completely forgot that this was unpublished and that we needed to show these important controls in the current paper – we apologize for this oversight! Please see our new supplemental figure (Figure 2 —figure supplement 2) that thoroughly addresses the impact of Auxin treatment on live oocytes. We observed no significant changes in either spindle length or pole to equator width (a ratio) when comparing untreated and auxin-treated oocytes.

4) There is a lot of ‚jumping’ between Figures 3 and 4 in the text (see paragraph “Dynein acts at spindle poles to promote pole formation and integrity”). The authors might re-think the order of call outs.

We have significantly restructured this section of the manuscript by moving the ASPM-1 and LIN-5 data earlier in the manuscript. We believe this has improved the clarity and general flow of the manuscript (since there is less jumping around), and we hope that the reviewer agrees.

5) Figure 4A-B shows key information. I think the panels are way too small, and I really encourage the authors to present these interesting findings in a better way.

We thank the reviewer for this excellent suggestion. We have restructured Figure 4, making the original Figure 4C its own new Figure (the new Figure 5), to give us room to increase the size of panels 4A and 4B. This also gave us enough room to add zoomed images to 4A, to emphasize key features from the video stills. We think that these changes make it easier for readers to evaluate our findings.

6) As for Figure 5 —figure supplement 1, I have a hard time to see the “labeling persisted on what appeared to be a singular disassembling RC between each segregating chromosome pair”. A clearer image should be presented.

We have switched out the previous image for a new image of SUMO labeling on mini anaphases; we think that the new image better shows the singular RC between separating chromosomes.

7) Line 384, add ‚monopolar’ to read: “resulted in a catastrophic breakdown of the monopolar spindle…”.

This is a good suggestion and we have made this change.

8) Lines 430-438, it is problematic that the authors are discussing the results of the presented manuscript in the context of unpublished data. This unpublished information should be documented somehow within this manuscript or deleted.

We understand the reviewer’s concern. Since the time of our original submission, we have posted a preprint to bioRxiv reporting the findings mentioned in the discussion . Therefore, we were able to add a reference to this preprint, which hopefully addresses this concern.

9) Line 479 and so forth (line 499, 529 etc.), did the authors use several ‚meiosis media’? In case only one buffer was used, change to ‚medium’. In any case, I would give the experimental conditions here and not fully rely on the details given in a reference.

We agree that it should be listed as “medium” so we have made this change. In addition, we have included the full recipe of this Meiosis Medium in the Methods section, as suggested.

10) Lines, 539-40, mention the exact auxin treatment protocols that were used within this study. Other, not applied protocols are irrelevant in the context of this paper.

We thank the reviewer for this suggestion. We have updated this section of the Methods (“Auxin treatment of *C. elegans*”) to only discuss the exact protocols presented in this study.

Reviewer #3:Cavin-Meza et al., have parsed out the importance of different motor proteins during oocyte spindle assembly in *C. elegans*. The authors developed an auxin-inducible dynein degradation system to remove dynein from the spindle. Using this system, they addressed the question of whether dynein is needed to maintain a bipolar spindle. With the depletion of dynein after spindle assembly, the spindles form parallel arrays but still maintain chromosomes in the middle. These results suggested that dynein was involved in pole focusing. To determine further explore the role of dynein in pole focusing, they depleted KLP-18, which normally gives monopolar spindle formation. With depletion of dynein, they find that the monopoles disappear and microtubule bundles form around the chromatin. Intriguingly, the microtubule bundles can allow chromosomes to segregate at anaphase. They then address the question of what motor protein is allowing anaphase onset. They knock out BMK-1 and find that bipolar spindle assembly and chromosome segregation no longer occurs. These results show that BMK-1 contributes to spindle assembly, but somewhat redundantly with other motors. Overall, the experiments were carefully planned and executed. The development of the degradable dynein is a useful tool for the fild. The novel finding that BMK-1 does contribute to spindle assembly is important for understanding the overlapping forces on the spindle. Furthermore, the results shows that there are differences between how the motors function between mitosis and meiosis, which is intriguing and important. Only an additional control experiments is needed.Overall, the manuscript details an important contribution of how the motor proteins are functioning together for spindle assembly. The manuscript is well-written, with clear rationales. I only have one concern:1) The aspm-1(RNAi) and lin-5(RNAi) and strains also contain the dynein degron background with dynein tagged. Although the controls show that without auxin, dynein is clearly working, sometimes the degron tags can cause a sensitized background due to a reduction in protein levels or activity. Although the phenotypes for the single depletions have been confirmed with other studies, the aspm-1(RNAi); klp1-18(RNAi) double mutant has not. Can the authors perform the same experiment with a strain without the tagged dynein?

We thank the reviewer for raising this important point. We agree that this would be a better version of the double RNAi experiment to perform. However, in response to comments from other reviewers, we have restructured the presentation of our ASPM-1 and LIN-5 experiments, and we have chosen to remove the double RNAi partial monopolar breakdown experiments from our paper. Thus, all the depletions in this revised manuscript are single depletion phenotypes that are corroborated by previous studies (such as Van der Voet et al., and Laband et al.,). Nevertheless, we appreciate the suggestion and we certainly would have performed these experiments in our TIR1 control strain if we were still including the experiments in question.

Reviewer #4:Cavin-Meza et al., investigated the role of dynein motors in meiotic spindle pole organization and spindle function using *C. elegans* oocytes as a model. They optimized the auxin-inducible degron system to degrade endogenous AID-tagged dynein acutely, rapidly, and more completely in oocytes. They find that dynein is required for integrity of bipolar spindle poles as well as spindle monopoles induced by depletion of the kinesin-12 motor KLP-18. They further show that in the absence of dynein motors, and functional spindle poles, microtubules are still able to organize into spindle-like structures via the action of chromosomes and the kinesin-5 motor BMK-1. They also found that chromosomes can still be segregated in anaphase-like motion in these oocytes.These results are consistent with many previous studies in mitotic and meiotic cells that investigated the function of dynein on spindle poles and chromosome segregation and showed spindle pole focusing function of this motor. Furthermore, it is known that upon removal of acentrosomal spindle pole components that assemble meiotic spindles in mouse oocytes, chromatin-mediated microtubule assembly can build spindles that can segregate chromosomes. This study is thus generally consistent with data from other models.The authors have optimized the AID-degron system beautifully and this gives them as well as the field new opportunities to study protein function in *C. elegans* oocytes in a much cleaner and more complete depletion background. The timescale in which they can deplete endogenous proteins is impressive.1. The authors previously published a paper where they found dynein inhibition causes chromosome segregation errors and that in this organism oocyte chromosomes are segregated independently of kinetochores in a pathway that involves dynein. While this is not in agreement with a more accepted model where inter-chromosomal microtubule arrays push chromosomes apart, it had so far provided an alternative model for oocyte chromosome segregation in the field. In this current manuscript, the authors show that dynein inhibition, presumably more complete and cleaner than other methods, in fact does not lead to any chromosome segregation errors. This is in direct conflict with their previously published work, but this major discrepancy is not addressed in the current work.

In response to this and a similar comment by Reviewer 1, we have expanded our Discussion to address this issue. The data presented in the current manuscript do not conflict with any of the data in our 2015 paper. Rather, new information has emerged in the years since we published our original paper that has changed our interpretation of the data in our 2015 paper, and has led to an updated view of chromosome segregation. Please see our response to comment #12 from Reviewer 1 for additional explanation.

2. The authors base their hypothesis on a previously published study in human oocytes reporting that spindles fail to maintain bipolarity and mis-segregate chromosomes. This work was performed on oocytes that were unhealthy and unsuitable for IVF treatment, and it is now clear that such atretic oocytes do not represent a general oocyte form. While the maternal age effect is indeed of clinical interest, given the strong interest in many areas of cell biology in understanding the fundamental principles of chromosome segregation in oocytes (of many species), it is not necessary to base experiments on observations made using human oocytes.

We did not intend for our introduction to give the impression that we are solely using Holubcova et al., 2015 as a basis for doing the experimentation in this paper; while we still think it is worthwhile to cite this study, we have updated the wording of the Introduction so it does not sound like this was the sole motivation for our study. We completely agree with this reviewer that the pursuit of fundamental knowledge about acentrosomal spindles and how they facilitate proper chromosome segregation is intriguing and motivating in of itself.

3. The contribution of chromatin-mediated mitotic and meiotic spindle assembly pathways was largely ignored in the discussion although these are likely to majorly contribute to miniature spindle assembly in dynein AID oocytes. Similarly, the contribution of dynein to spindle pole focusing in other systems needs more coverage.

We thank the reviewer for these great suggestions. We agree that chromatin-mediated microtubule nucleation may play a role in the formation of the miniature anaphase spindles. We have now added text to the Discussion to raise this interesting idea and we have cited several reviews on this topic. We also appreciate the suggestion to add information about dynein’s role in spindle pole focusing in other systems. Most of our references involving pole focusing driven by dynein were originally centered in mitosis, so we have added some additional background on dynein’s roles in acentrosomal spindle assembly to our Introduction, citing studies using *Xenopus* egg extract (Heald et al., 1996), *Xenopus oocytes* (Becker et al., 2003), and mouse oocytes (Luksza et al., 2013, Clift and Schuh 2015).

4. While the images are of high quality and very accessible, the lack of appropriate image analysis and quantification means the manuscript is largely descriptive. It would be important to support several conclusions made throughout the paper with robust quantification and statistical analyses.

We have addressed this concern in a number of ways (many of these changes are also detailed in our previous responses to the other reviewers, who made specific suggestions along these lines). To better support our conclusions, we have made efforts to provide more rigorous quantification of numerous aspects of our data. First, we performed new pole splaying quantifications. Instead of judging whether a pole was “focused” or “splayed” by eye (as we did in the original submission), we measured the widths of the poles and of the center of the spindle; we then calculated a ratio of the pole to equator width as a measurement of how splayed the poles are (this is shown in Figure 2B). This new analysis confirmed the finding that we reported in our original manuscript – that the poles are substantially splayed upon auxin treatment. Additionally, we have added rigorous controls to demonstrate that the effects we see are not caused by auxin treatment (Figure 2 —figure supplement 2). We dissected oocytes lacking degron-tagged dynein into auxin and then imaged spindles over time, demonstrating that neither the spindle length nor the pole-to-equator ratio changed. Finally, we now report the number of images analyzed for all figures, we have added quantitative information into the manuscript when discussing results, and we have tried to emphasize the more quantitative aspects of our paper in our revised manuscript, such as our measurements of chromosome segregation rates and distance traces in live imaging.

5. While the writing is quite accessible to readers working on the *C. elegans* model, it is much less clear for those who work on other models. It would be very helpful to re-write technical bits such as 'embryos dissected into auxin'. It is also important to include citations from other models when summarizing what is known about certain spindle features in oocytes. There is generally ambiguity in the writing that leaves the reader wondering about the precise point that is being made. E.g. microtubule bundles were able to remain relatively aligned' – this and several other statements in the paper are very descriptive and without quantitative image analysis to back them up, it is difficult to grasp what they mean.

We definitely want to make the manuscript accessible to readers working on different models so we appreciate this feedback. In response, we have tried to edit some of the worm-specific language to make it less confusing, we have added a schematic to Figure 1A to better explain our auxin treatment methods (“long-term”, “short-term”, and “acute”), and we have changed the figures and text to be consistent when referring to these methods. We have also gone through the manuscript and have tried to remove ambiguity (e.g. removing phrases such as “relatively aligned” and replacing them with more precise descriptions). We think that these changes will help understanding. In addition, we understand the desire for additional references detailing work in other systems. We were originally very brief in our introduction in an effort to be succinct. However, in response to this concern, we have attempted to elaborate further on important points (as mentioned in comment #3) and we hope that this reviewer will now feel that we have provided adequate information for all readers to find the results of this study clear and accessible.

This manuscript can be significantly improved by providing a detailed discussion concerning the conflict between the data within and the lab's previously published work that shows dynein inhibition causes chromosome segregation errors. Furthermore, robust image quantification and statistical analyses are needed to make the paper much less descriptive cell biology than it is now. It would also be good to include some controls to show that 4 hour treatment of non-AID strain wild-type oocytes has no effect on several aspects of spindle assembly and function studied here.

We have now included a discussion of our previous work (see the response to comment #1 above) and more quantitative analysis (see the response to comment #4 above). We thank the reviewer for the excellent suggestion of an additional control experiment, and we agree that it is important to show that long-term AID (4 hours on auxin-containing plates) does not have adverse effects on acentrosomal spindle assembly. We have now included a new supplemental figure (Figure 1 —figure supplement 1) reporting this analysis. We repeated our germline counting to assess proper spindle formation in a strain that contains all the same fluorophores and transgenes, but that is lacking the degron::GFP tag on DHC-1. We found no discernable difference between untreated and treated oocytes across a large sample size.